# Deletion of skeletal muscle *Akt1/2* causes osteosarcopenia and reduces lifespan in mice

Takayoshi Sasako [1,2], Toshihiro Umehara[1], Kotaro Soeda [1,2], Kazuma Kaneko[1], Miho Suzuki[1], Naoki Kobayashi[2], Yukiko Okazaki[1], Miwa Tamura-Nakano[3], Tomoki Chiba[4], Domenico Accili [5], C. Ronald Kahn [6], Tetsuo Noda[7], Hiroshi Asahara[4], Toshimasa Yamauchi[1], Takashi Kadowaki [1,8] ✉ & Kohjiro Ueki[2,9] ✉

Aging is considered to be accelerated by insulin signaling in lower organisms, but it remained unclear whether this could hold true for mammals. Here we show that mice with skeletal muscle-specific double knockout of *Akt1/2*, key downstream molecules of insulin signaling, serve as a model of premature sarcopenia with insulin resistance. The knockout mice exhibit a progressive reduction in skeletal muscle mass, impairment of motor function and systemic insulin sensitivity. They also show osteopenia, and reduced lifespan largely due to death from debilitation on normal chow and death from tumor on high-fat diet. These phenotypes are almost reversed by additional knocking out of *Foxo1/4*, but only partially by additional knocking out of *Tsc2* to activate the mTOR pathway. Overall, our data suggest that, unlike in lower organisms, suppression of Akt activity in skeletal muscle of mammals associated with insulin resistance and aging could accelerate osteosarcopenia and consequently reduce lifespan.

Findings from lower organisms have shown that insulin and insulin-like growth factor-1 (IGF-1) signaling accelerates aging[1,2], but it remains unclear whether this could also be entirely true for mammals. Indeed, impaired action of insulin/IGF-1 signaling is counted among the potential mechanisms underlying the development of sarcopenia, a major phenomenon associated with aging, which is characterized by decreased mass and impaired function of skeletal muscle[3–6]. Sarcopenia is known to be promoted in patients with diabetes in which insulin action is impaired but likely to be ameliorated by treatment with insulin sensitizers[7]. Moreover, restoration, not inhibition, of IGF-1 was assumed to represent a promising therapeutic approach to sarcopenia[5].

Sarcopenia, in its turn, is thought to be one of the main causes of insulin resistance in older subjects because glucose uptake in an absorptive state is attributed in large part to skeletal muscle, while it is known to be impaired in patients with diabetes, resulting in glucose intolerance and hyperglycemia[8,9].

Most transducers of the signaling cascade are shared by insulin and IGF-1: insulin receptor (IR) and IGF-1 receptor (IGF1R) phosphorylate insulin receptor substrates (IRSs), which activate phosphoinositide 3-kinase and its downstream Akt, resulting in inactivation of FoxOs involved in protein degradation, as well as in activation of mTOR involved in protein synthesis[10,11]. Thus, it is assumed that not only

[1]Department of Diabetes and Metabolic Diseases, Graduate School of Medicine, The University of Tokyo, Tokyo, Japan. [2]Department of Molecular Diabetic Medicine, Diabetes Research Center, National Center for Global Health and Medicine, Tokyo, Japan. [3]Communal Laboratory, Research Institute, National Center for Global Health and Medicine, Tokyo, Japan. [4]Department of Systems BioMedicine, Tokyo Medical and Dental University, Tokyo, Japan. [5]Columbia University College of Physicians & Surgeons, Department of Medicine, New York, NY, USA. [6]Joslin Diabetes Center, Harvard Medical School, Boston, MA, USA. [7]Department of Cell Biology, Cancer Institute, Japanese Foundation of Cancer Research, Tokyo, Japan. [8]Toranomon Hospital, Tokyo, Japan. [9]Department of Molecular Diabetetology, Graduate School of Medicine, The University of Tokyo, Tokyo, Japan. ✉e-mail: t-kadowaki@toranomon.kkr.or.jp; uekik@ri.ncgm.go.jp

insulin signaling but also IGF-1 signaling are impaired in patients with insulin resistance.

In order to examine the roles of the insulin/IGF-1 signaling in skeletal muscle, various model mice have been generated and analyzed[12]. It was implied that both insulin and IGF-1 are required in the maintenance of muscle mass[13,14] and that both activation of FoxOs and inactivation of mTOR negatively regulate muscle mass[15–17], although there remains some room for controversy[14,18–21]. In these models, impairment of insulin/IGF-1 signaling did not necessarily lead to systemic insulin resistance. Moreover, the phenotypes in some models were not fully examined due to death while young, and those in others which were viable were examined mainly in young animals. Thus a good mouse model remained to be established in which normally developed skeletal muscle gradually becomes sarcopenic, accompanied by insulin resistance as they grow older, and the course observed in humans is closely replicated.

Akt is a key downstream kinase, and of its three isoforms, mainly Akt1 and Akt2 are expressed in skeletal muscle[22]. Systemic deletion of *Akt1*, *Akt2*, and *Akt1/2* resulted in mild growth retardation, insulin resistance, and neonatal lethality due to severe growth retardation, respectively[23–25]. However, a model of Akt-deleted skeletal muscle remained to be established.

In this study, we generated skeletal muscle-specific *Akt1/2* double-knockout mice and demonstrated that the knockout mice exhibited premature sarcopenic phenotypes, accompanied by systemic insulin resistance. They also showed osteopenia and reduced lifespan, suggesting a crucial role for skeletal muscle Akt in the regulation of aging of skeletal muscle and lifespan.

## Results

### Generation of skeletal muscle-specific *Akt1/2* knockout mice
To elucidate the effects of insulin signaling on skeletal muscle mass, we first investigated two mouse models, i.e., mice treated with streptozotocin, a model of insulin deficiency, and *db/db* mice, a model of severe insulin resistance[26]. Indeed, the two models shared significantly decreased fast-twitch muscle (represented by extensor digitorum longus [EDL] and gastrocnemius) mass, non-altered slow-twitch muscle (represented by soleus) mass, and relatively decreased type 2 myofibers characterizing fast-twitch muscle (Supplementary Fig. 1a–d). Thus impaired insulin action was thought likely to result in dys-regulation of skeletal muscle mass and myofiber type mimicking the characteristics of sarcopenia in humans[27].

Then we examined how insulin signaling in skeletal muscle was affected by aging to explore the impact of insulin signaling on the development of sarcopenia. When the mice were treated with insulin, IR and IRS, upstream molecules of skeletal muscle were phosphorylated to a similar extent in younger and older wild-type mice alike, while phosphorylation of Akt, a key downstream kinase, was suppressed in older mice alone (Supplementary Fig. 1e).

These data urged us to hypothesize that sarcopenia could be promoted by insulin resistance in skeletal muscle associated with aging, especially at the downstream level, and as a model of such insulin resistance, we generated mice with tissue-specific deletion of *Akt1* and *Akt2*, the two major isoforms of Akt in skeletal muscle[22]. We selected mice with Cre recombinase knocked in the downstream of the myosin light-chain (*Mlc1f*) promoter[28], to ensure skeletal muscle-specific expression of the recombinase, followed by crossing with *Akt1/2*-floxed mice.

In the skeletal muscle-specific *Akt1/2* double-knockout (mAktDKO) mice, the Akt proteins were successfully down-regulated in skeletal muscle, but not in other tissues, including the heart (Fig. 1a, b & Supplementary Fig. 2a, b). It was also confirmed that when the mAktDKO mice were treated with insulin, phosphorylation of Akt was almost blunted, and phosphorylation of downstream molecules involved in protein synthesis, protein degradation, and glucose uptake were also suppressed in fast-twitch muscle (Fig. 1c & Supplementary Fig. 2c). Knocking out of *Akt1/2* was less sufficient in slow-twitch muscle and phosphorylation of the downstream molecules were not suppressed (Fig. 1a, d & Supplementary Fig. 2d).

### Sarcopenic phenotypes of muscle-specific *Akt1/2* knockout mice
Then we followed up on the body weight of the mAktDKO mice until they were 100 weeks old. They weighed significantly lighter from the age of 8 weeks onward than the control mice (Fig. 2a). Given that the mAktDKO and control mice were similar in body length and food intake (Supplementary Fig. 3a, b) and did not differ in fat mass (Fig. 2b & Supplementary Fig. 3c), the difference in body weight was thought likely to be due to reductions in lean body mass in the mAktDKO mice.

Consistently, the fast-twitch muscle mass in the mAktDKO mice was almost half that of the control mice, whereas their slow-twitch muscle mass and white adipose tissue mass were not reduced or reduced only slightly compared to those in the control mice (Fig. 2c & Supplementary Fig. 3d). Histological analysis revealed that the myofibers with a smaller cross-sectional area were increased (Fig. 2d) in the fast-twitch muscle of the mAktDKO mice compared to those in the control mice.

In parallel with a reduction in fast-twitch muscle mass characterized by spontaneous power, grip strength was significantly weaker in the older mAktDKO mice than that in the control mice (Fig. 3a). Besides, the exercise duration was markedly shorter in the older mAktDKO mice (Fig. 3b), and consistently, the ATP content in skeletal muscle of the mAktDKO mice after exercise was decreased compared to that of the control mice, although it was not even after extended endurance exercise in the younger mAktDKO mice (Fig. 3c). No sign of heart failure was observed in terms of heart weight and expression of diuretic peptides (Supplementary Fig. 3e, f).

In addition to motor function, glucose uptake is another important function of skeletal muscle. Indeed, insulin-induced glucose uptake examined ex vivo was impaired in the fast-twitch muscle of the older mAktDKO mice (Fig. 3d, e). Consistently, systemic insulin sensitivity examined by insulin tolerance test was apparent in the older mAktDKO mice, and marked insulin resistance and glucose intolerance were observed at the age of 100 weeks, although blood glucose in an *ad libitum*-fed state was comparable to that in the control mice (Fig. 3f–h & Supplementary Fig. 3g).

Therefore the mAktDKO mice may represent a good model to elucidate the pathogenesis and mechanisms of sarcopenia, in which loss of mass and dys-function of skeletal muscle were induced and accelerated with aging, accompanied by insulin resistance and glucose intolerance, in contrast to the skeletal muscle-specific *Akt1* or *Akt2* single-knockout mice, which did not exhibit reduced body weight (Supplementary Fig. 3h, i).

### Mechanisms underlying the sarcopenic phenotypes of the muscle-specific *Akt1/2* knockout mice
Next we explored mechanisms underlying the sarcopenic phenotypes of the mAktDKO mice. Genes highlighting fast-twitch muscle, type 2 myofibers (represented by *Myh4*) and glycolysis (*Pfkm*), were down-regulated in the fast-twitch muscle of the older mAktDKO mice (Fig. 4a & Supplementary Fig. 4a), while that encoding type 1 myofibers (*Myh7*) were not (Fig. 4a). ATPase staining confirmed that type 2 myofibers were relatively decreased in the mAktDKO mice (Fig. 4b).

The other mechanism responsible for ATP production is oxidative phosphorylation, and the genes involved in mitochondria biogenesis (*Tfam*, *Mfn2*) were down-regulated (Fig. 4a & Supplementary Fig. 4a), followed by a decrease in mitochondria DNA copy number (Fig. 4c). Moreover, those encoding electron transport chain components (*Ndfua8*), which are known to be regulated by insulin signaling[29], as well as sirtuin 3, which regulates acetylation of the components, were down-regulated (Fig. 4a & Supplementary Fig. 4a). These changes in

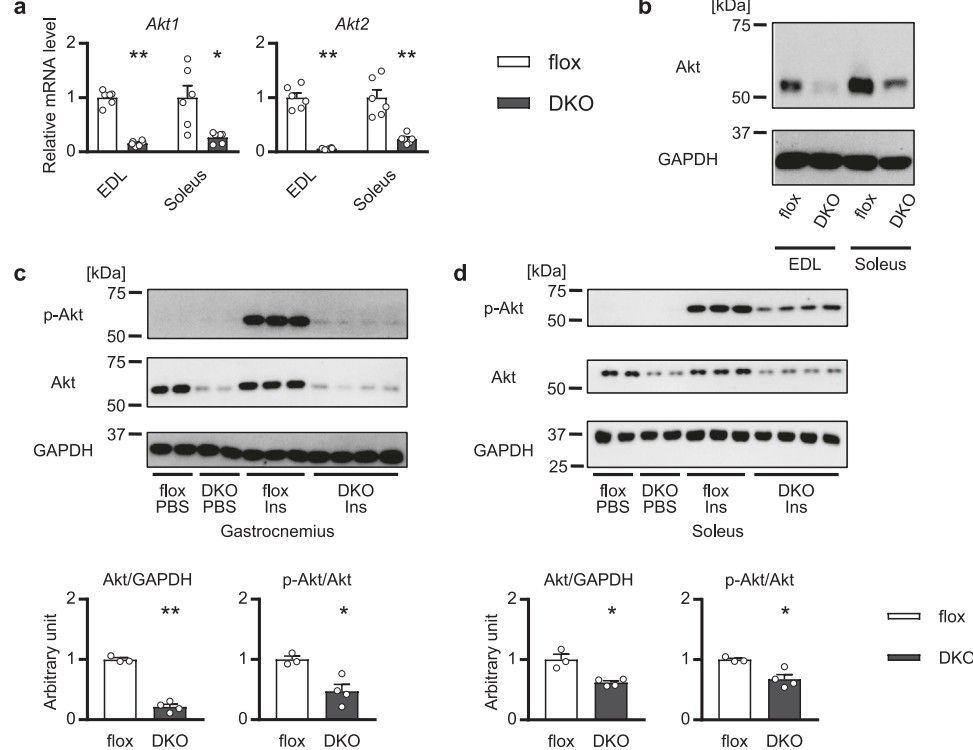

**Fig. 1 | Generation of skeletal muscle-specific *Akt1/2* knockout mice.** Expression of Akt in EDL and soleus of the mAktDKO mice at the age of 8 weeks analyzed by, **a** RT-PCR (*n* = 6 mice per group, as shown by the number of data points on the graph, same as below), and **b** western blotting. **c**, **d** Insulin signaling in **c** gastrocnemius, and **d** soleus, at the age of 8 weeks after treatment with insulin for 10 minutes analyzed by western blotting (*n* = 3 or 4 mice). Values of the data are expressed as mean ± SEM. \*P < 0.05, \*\*P < 0.01. Unpaired 2-tailed *t*-test was used for assessment, and the exact P values are provided in Supplementary Data 2. Source data are provided as a Source Data file.

mitochondria-related gene expression were found to highlight the results of transcriptome analysis (GEO accession: GSE199074) of the fast-twitch muscle of the aged mAktDKO mice (Supplementary Data 1). Consistently, electron microscopic analysis revealed morphological abnormalities of mitochondria characterized by mitochondrial swelling and low electron density (Fig. 4d, e & Supplementary Fig. 5a). AMP-activated protein kinase α (AMPKα) was activated even after extended endurance exercise in the younger mAktDKO mice, but it was not in the older mAktDKO mice (Supplementary Fig. 5b, c), in accordance with the changes in ATP content (Fig. 3c).

We also studied mitophagy as one of the mechanisms responsible for the elimination of such abnormal mitochondria. Autophagy-related genes (*Atg4a*) were down-regulated (Fig. 4a & Supplementary Fig. 4a), but conversely, LC3B-II protein, a component of the autophagosome, was accumulated (Fig. 4f). LC3B-II protein was increased by lysosomal inhibition by administration of leupeptin in control floxed mice but was not in the mAktDKO mice, suggesting impairment in autophagy flux. Moreover, LC3B-II protein was detected in the mitochondrial fraction, which was increased in the mAktDKO mice (Supplementary Fig. 5d, e). In electron microscopic analysis, mitophagy-like events involving autophagosomes containing abnormal mitochondria inside were significantly increased, with electron-dense aggregates left nearby (Fig. 4e & Supplementary Fig. 5a), suggesting that mitophagy failure could be induced in the fast-twitch muscle of the mAktDKO mice. Genes involved in reactive oxygen removal (*Sod2*), as well as in fatty acid oxidation, were also down-regulated (Fig. 4a & Supplementary Fig. 4a). Of note, genes involved in proteasome-mediated protein degradation, which are known to be positively regulated by FoxO3[30], and pathways involved in proteasome and ubiquitination were shown to be unaffected by RT-PCR and transcriptome analysis (GSE199074), respectively (Supplementary Fig. 4b & Supplementary Data 1), suggesting that the reduced muscle mass observed in the mAktDKO mice

could not be largely attributed to increased protein degradation by the proteasome pathway.

Given that mitochondria, oxidative stress, and proteostasis are closely associated with aging[2], these data implied accelerated aging in the mAktDKO mice, and indeed, sirtuin 1 was down-regulated (Supplementary Fig. 4a), whereas an aging marker (*Ink4a*) was highly up-regulated in the fast-twitch muscle of the mAktDKO mice (Fig. 4a), associated with enhanced staining of the senescence-associated beta-galactosidase (Fig. 4g).

Notably, these changes were less evident in the slow-twitch muscle of the mAktDKO mice (Supplementary Fig. 5f, g).

## Skeletal muscle-specific knockout of *Akt1/2* causes osteopenia and reduced lifespan

Then we hypothesized that accelerated aging of skeletal muscle might affect aging of other tissues. Because skeletal muscle and bone are closely associated with each other, we focused on osteoporosis, one of the phenomena associated with aging in bone. Indeed, the mAktDKO mice had a significantly lower femur weight than the control mice (Fig. 5a). Computed tomography (CT) scanning of lower leg revealed decreased bone mineral density (BMD) in spongy bone compared to the control mice (Fig. 5b), and the star volume, a good parameter of osteoporosis, was larger in the femur of the mAktDKO mice on micro-CT scanning (Fig. 5c & Supplementary Fig. 6a).

Bone morphometric analysis revealed that the osteoblast surface showed a non-significant decrease, the osteoclast surface was not affected, and the bone formation rate was significantly lower in the mAktDKO mice, compared to the control mice (Fig. 5d). Mineralization was also shown to be impaired by von Kossa staining (Fig. 5e), suggesting that osteopenia could be induced in the mAktDKO mice via impaired osteogenesis, rather than via enhanced osteolysis. It was also confirmed that the gene expression of myokines that could modulate

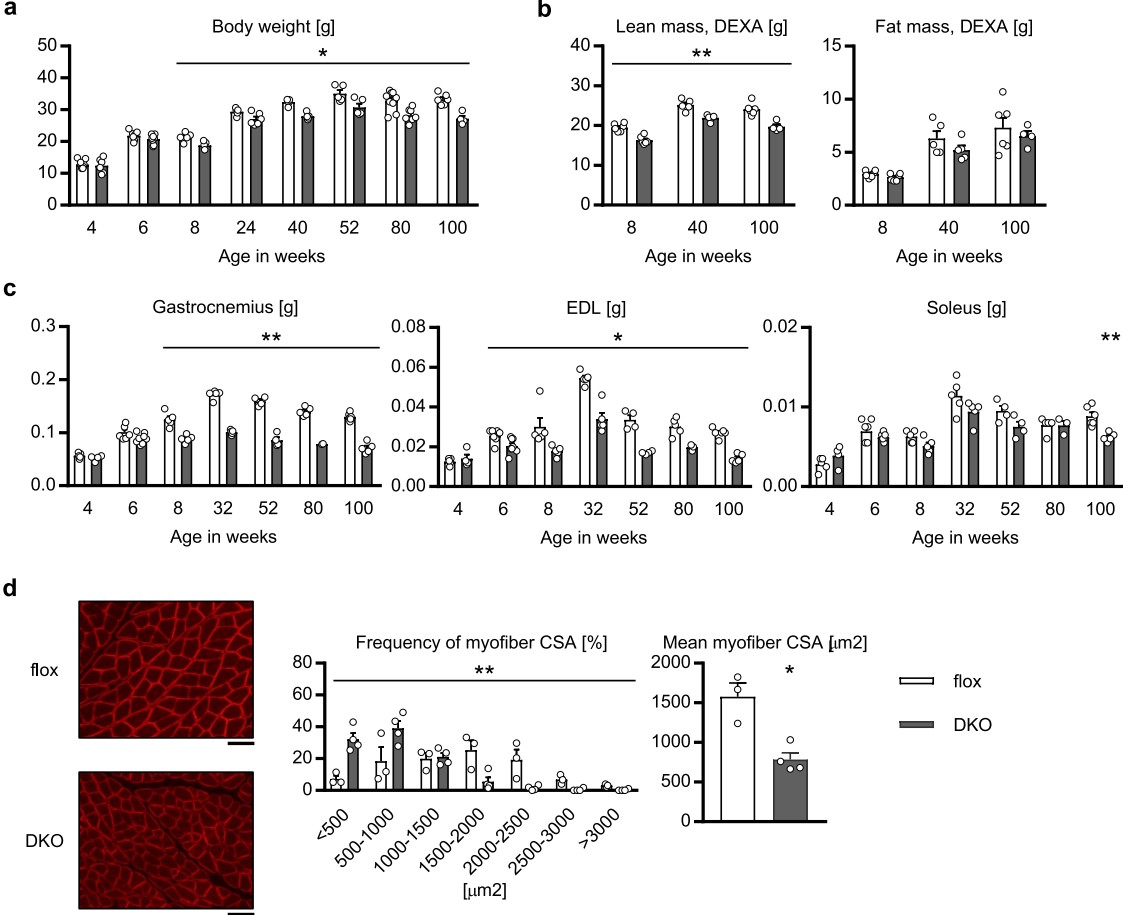

**Fig. 2 | Body and muscle weight of the mAktDKO mice. a** Body weight ($n$ = 4, 5, 6, 7, 8 or 9 mice per group, as shown by the number of data points on the graph, same as below) at the age of the indicated weeks. **b** Lean body mass and fat mass analyzed by dualenergy X-ray absorptiometry (DEXA) at the age of the indicated weeks ($n$ = 4, 5, or 6 mice). **c** Skeletal muscle weight at the age of the indicated weeks ($n$ = 3, 4, 5, 6, 7, or 8 mice). **d** Immunostaining of EDL with anti-dystrophin antibody and analysis of cross-sectional area (CSA) of myofibers at the age of 80 weeks ($n$ = 3 or 4 mice). Values of the data are expressed as mean ± SEM. *$P$ < 0.05, **$P$ < 0.01. Unpaired 2-tailed $t$-test (**a**–**c**) and chi-square test with Yates' correction **d** were used for assessment, and the exact $P$ values are provided in Supplementary Data 2. Scale bars: 50 μm. Source data are provided as a Source Data file.

osteogenesis was affected, while the expression of the local form of *Igf1* was not (Supplementary Fig. 4a).

Of note, osteopenia was observed even in the mAktDKO mice at the age of 6 weeks (Supplementary Fig. 6b), when loss of skeletal muscle mass was not so evident. Calvarial cells were isolated and their autonomous differentiation to osteoblasts was shown to be intact (Supplementary Fig. 6c–e).

Then the mAktDKO mice were followed up on their lifespan, which turned out to be significantly reduced (Fig. 5f). We noticed that many of the mAktDKO mice died with a skinny appearance (Fig. 5g & Supplementary Fig. 6f), which prompted us to analyze their dead body weight. It was indeed shown to be significantly lower, with the mean weight being even below 10 percentile (-20 g) of the control mice (Fig. 5h). We herein defined deaths from debilitation as those with no macroscopic findings and with a dead body weight less than 20 g. Although more than half of the control mice died from tumor, half of the mAktDKO mice died from debilitation, with this difference shown to be significant (Fig. 5i). The cumulative incidence of death from debilitation showed an apparent separation, but that of death from tumor did not (Supplementary Fig. 6g).

Given that all of these phenotypes were observed in male mice, we also examined whether they were shared with female mice. Those in skeletal muscle observed in male mAktDKO mice were also shared (Supplementary Fig. 7a–c), but the systemic phenotypes concerning body weight, systemic insulin resistance, and lifespan, as well as those

in bone, were not shared by the female mAktDKO mice (Supplementary Fig. 7d–k). Therefore it was suggested that the impaired activity of Akt in skeletal muscle could be essential to the maintenance of volume and function of the skeletal muscle itself even in females, but its impact on other tissues could be smaller in females compared to that in males.

## Muscle-specific *Akt1/2* knockout mice in under- and over-nutrition

Skeletal muscle is also known as a tissue to cope with fasting by supplying alanine as a substrate for gluconeogenesis in the liver. When fasted for 24 h, the mAktDKO mice showed significantly lower blood glucose levels (Fig. 6a), although their body weight was lost in a manner proportional to that in the control mice (Supplementary Fig. 8a). Serum alanine levels were significantly lower (Fig. 6b), and serum ketone body levels were conversely elevated (Fig. 6c), while serum fatty acid levels were not altered (Supplementary Fig. 8b), in the mAktDKO mice, compared to the control mice.

Then we examined effects of caloric restriction for weeks on the phenotypes of the mAktDKO mice. Caloric restriction by 40% started at the age of 90 weeks led to elevation of serum ketone body levels (Supplementary Fig. 8c), but exercise duration was not improved in the mAktDKO mice (Fig. 6d). Although the grip strength of the mAktDKO mice was improved proportionally to that in the control mice (Supplementary Fig. 8d), their body weight was further reduced (Supplementary Fig. 8e), and eventually their lifespan was markedly

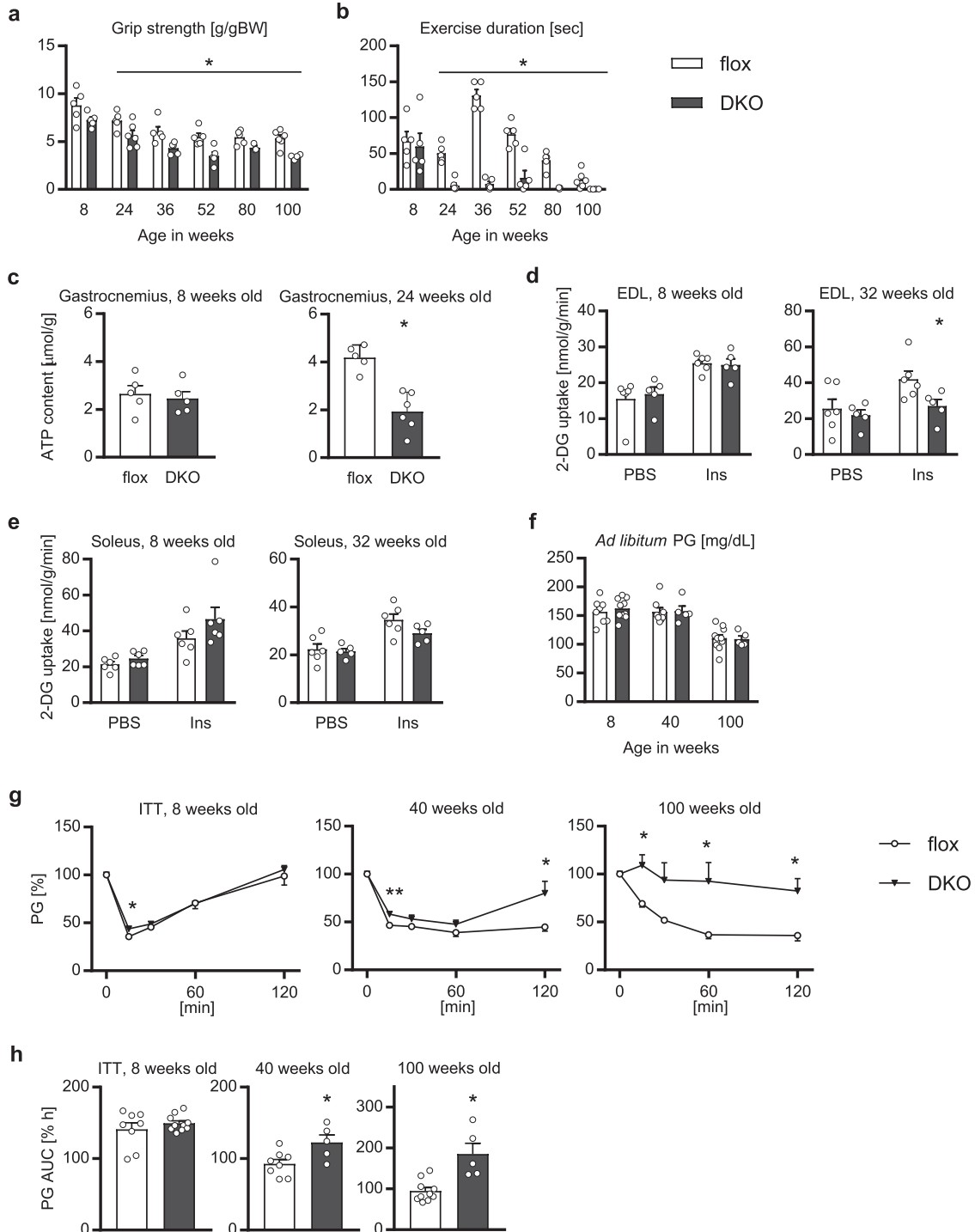

**Fig. 3 | Skeletal muscle-associated phenotypes of the mAktDKO mice. a** Grip strength and **b** exercise duration, of the mAktDKO mice at the age of the indicated weeks ($n = 3, 4, 5$ or 6 mice per group, as shown by the number of data points on the graph, same as below). **c** ATP contents of gastrocnemius after exercise for 60 minutes at the age of 8 weeks ($n = 5$ mice) and after exercise for 15 minutes at the age of 24 weeks ($n = 5$ or 6 mice). **d, e** Insulin-dependent glucose uptake in, **d** EDL and **e** soleus, analyzed ex vivo using 2-DG at the age of the indicated weeks ($n = 5$ or 6 mice). **f–h** *Ad libitum* plasma glucose (PG) before insulin challenge (**f**), **g** relative PG after insulin challenge, and **h** area under the curve (AUC) of relative PG after insulin challenge, in insulin tolerance test (ITT), after intraperitoneal injection of human regular insulin (0.75 U/kg BW, 1.0 U/kg BW, and 1.5 U/kg BW, at the age of 8, 40, and 100 weeks, respectively) ($n = 8$ or 10 mice at the age of 8 weeks, $n = 5$ or 8 mice at the age of 40 weeks, and $n = 5$ or 10 mice at the age of 100 weeks). Values of the data are expressed as mean ± SEM. *$P < 0.05$, **$P < 0.01$. Unpaired 2-tailed $t$-test was used for assessment, and the exact $P$ values are provided in Supplementary Data 2. Source data are provided as a Source Data file.

reduced (Fig. 6e), with none dying from tumor, compared with the control mice.

We also generated a model of sarcopenia in over-nutrition by feeding the mAktDKO mice with high-fat diet. The phenotypes observed in the skeletal muscle of the mAktDKO mice fed with normal chow were still observed in the mAktDKO mice fed with high-fat diet, as well as those seen in bone, despite un-altered food intake (Fig. 6f–i & Supplementary Fig. 8f). Body weight and systemic insulin resistance, however, showed no difference between the mAktDKO mice and the control mice, although adiposity was increased in the

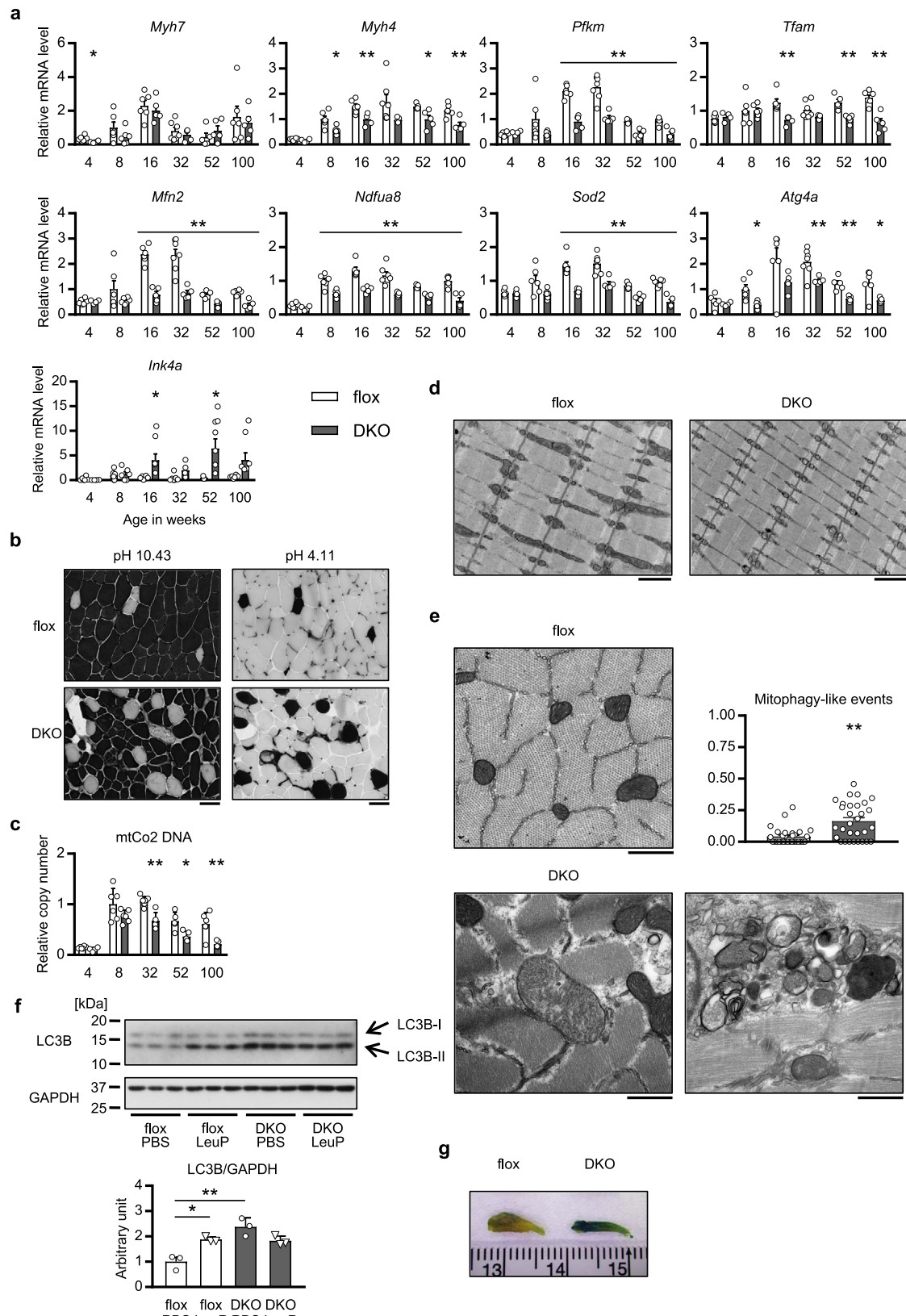

mAktDKO mice compared to the control mice (Fig. 6j, k & Supplementary Fig. 8g–i). The gene expression profile of epididymal white adipose tissue showed no particular differences between groups (Supplementary Fig. 8j). The mAktDKO mice fed with high-fat diet still exhibited reduced lifespan (Fig. 6l), many of which died from tumor, not from debilitation, just as the control mice did (Fig. 6m, n &

Supplementary Fig. 8k, l). Given that on normal chow diet, the mAktDKO mice did not show increased death from tumor, it was hypothesized that under the circumstances, such as high-fat diet, which promote tumor development, the mAktDKO mice could accelerate tumor growth. To test this hypothesis, we subcutaneously transplanted melanoma cells to the mice, and found that the cell

**Fig. 4 | Mechanisms underlying the phenotypes of the mAktDKO mice. a** Gene expression, **c** DNA copy number, of EDL of the mAktDKO mice at the age of the indicated weeks analyzed by RT-PCR ($n = 4, 5, 6,$ or 7 mice per group, as shown by the number of data points on the graph, same as below). **b** ATPase staining of EDL at the age of 52 weeks. Scale bars: 50 μm. **d, e** Electron microscopic image(s) of, horizontal sectioning (**d**), and vertical sectioning (**e**), of EDL at the age of 60 weeks. **e** The percentage of mitophagy-like events were quantified ($n = 30$ random fields from 3 mice). Scale bars: **d** 1 μm, and **e** 500 nm. **f** An autophagy-related protein in gastrocnemius at the age of 40 weeks 4 h after intraperitoneal administration of leupeptin at the dose of 40 mg/kg of BW in an *ad libitum*-fed state analyzed by western blotting ($n = 3$ mice per group). **g** Macroscopic image of SA-BG staining of EDL at the age of 80 weeks. Scale graduation: 1 mm. Values of the data are expressed as mean ± SEM. *$P < 0.05$, **$P < 0.01$. Unpaired 2-tailed *t*-test (**a, c, e**) and one-way ANOVA (**f**) were used for assessment, and the exact *P* values are provided in Supplementary Data 2. Source data are provided as a Source Data file.

growth was faster in the mAktDKO mice compared to the control mice (Supplementary Fig. 8m, n).

Therefore the mouse model of sarcopenia showed reduced lifespan in both under- and over-nutrition, as well as in normo-nutrition.

### Ablation of FoxOs rescued the phenotypes of the muscle-specific *Akt1/2* knockout mice

Lastly, we explored a key molecule which could account for the phonotypes induced by knocking out of *Akt1/2* in skeletal muscle. Of the molecules downstream of Akt, we focused on FoxOs and mTOR, which are mainly involved in protein degradation and protein synthesis, respectively[10,11].

First, we tried to inactivate the FoxO pathway, which was activated by knocking out of *Akt1/2*. We focused on Foxo1 and Foxo4, major isoforms of FoxOs in skeletal muscle which were recently reported to be major FoxOs downstream of Akt[31], and generated skeletal-specific *Akt1/Akt2/Foxo1/Foxo4* quadruple-knockout (mAkt/FoxoQKO) mice (Supplementary Fig. 9a, b). Fast-twitch muscle mass, body weight, lean mass, and motor function were all preserved in the mAkt/FoxoQKO mice, without altering body length, food intake, and adiposity (Fig. 7a–e & Supplementary Fig. 9c–f).

The changes in gene expression in the fast-twitch muscle of the mAktDKO mice, as well as that in mitochondria copy number, were restored in the mAkt/FoxoQKO mice, with no morphologically abnormal mitochondria observed (Fig. 7f–h & Supplementary Fig. 9a), while systemic insulin resistance was rescued only partially (Fig. 7i & Supplementary Fig. 9g). Femur weight was also preserved in the mAkt/FoxoQKO mice, with the findings suggestive of osteopenia shown in the mAktDKO mice also diminished (Fig. 7j, k & Supplementary Fig. 9h).

The mAkt/FoxoQKO mice were also shown to have an equivalent lifespan to that in the control mice, with no reduction in dead body weight or an increase in death from debilitation (Fig. 7l–n), suggesting that inhibition of FoxO activity might be a promising strategy for sarcopenia, as well as for osteopenia and reduced lifespan. Indeed, administration of a FoxO inhibitor, AS1842856[32], for 4 weeks partially reversed the reduced fast-twitch muscle mass of the aged mAktDKO mice, whereas it did not affect slow-twitch muscle mass (Fig. 7o & Supplementary Fig. 9i, j).

We also attempted to activate the mTOR pathway, which was inactivated by knocking out of *Akt1/2*. Akt is known to phosphorylate and inactivate TSC2, which, in turn, suppresses activation of mTOR[10,11], and thus we generated skeletal muscle-specific *Akt1/Akt2/Tsc2* triple-knockout (mAkt/TscTKO) mice (Supplementary Fig. 10a, b). The mAkt/TscTKO mice failed to rescue the reduced body weight and reduced fast-twitch muscle weight observed in the mAktDKO mice (Fig. 8a, b); still they had grip strength restored (Fig. 8c), and their exercise duration was partially preserved (Fig. 8d).

The suppressed expression of genes involved in glycolysis and mitochondria observed in the mAktDKO mice was rescued in the mAkt/TscTKO mice (Fig. 8e & Supplementary Fig. 10a). Type 2 myofibers were still down-regulated, while type 1 myofibers which characterize slow-twitch muscle and are abundant in mitochondria were up-regulated (Fig. 8e). They also showed an increased mitochondria copy number (Fig. 8f), despite still exhibiting morphological abnormalities (Fig. 8g). Moreover, the expression of an anti-oxidant enzyme was still down-regulated, and the expression of an aging marker was highly up-regulated in these mice (Fig. 8e). Femur weight reduction and osteopenic changes on micro-CT scanning were not rescued (Fig. 8h, i & Supplementary Fig. 10c).

Indeed, the mAkt/TscTKO mice exhibited shorter body length, decreased adiposity, lower blood glucose in an *ad libitum*-fed state, and enhanced systemic insulin sensitivity, despite unaltered food intake (Supplementary Fig. 10d–h), and consistently, *Igf1* was down-regulated in both skeletal muscle and the liver (Supplementary Fig. 10a, i). Such phenotypes of growth retardation were observed as early as 4 weeks of age (Supplementary Fig. 10j–l). Premature death within a month after weaning was observed in a proportion of the mAkt/TscTKO mice (Fig. 8j), and even the survivors for 8 weeks after birth exhibited reduced lifespan (Fig. 8j) with their dead body weight < 20 g (Supplementary Fig. 10m).

Overall, activation of the mTOR pathway did not fully rescue the sarcopenic phenotypes due to skeletal muscle-specific deletion of *Akt1/2*. However, glycolysis, electron transport chain components, and autophagy were shown to be regulated by the mTOR pathway, as well as by the FoxO pathway. Type 2 myofibers and anti-oxidative capacity were regulated solely by the FoxO pathway, whereas type 1 myofibers and the quantity of mitochondria were predominantly regulated by the mTOR pathway.

## Discussion

Insulin/IGF-1 signaling has been assumed to accelerate aging based on findings mainly in lower organisms. Here we show, rather, that impaired insulin/IGF-1 signaling in skeletal muscle in mice not only accelerates aging of skeletal muscle itself but also reduces lifespan.

Sarcopenia is one of the phenomena associated with aging in skeletal muscle, and it has been suggested that, of the potential mechanisms of the development of sarcopenia, insulin resistance and insufficient IGF-1 action could be major contributors[5,33]. Indeed, we observed suppressed activity of Akt, a key downstream transducer of insulin/IGF-1 signaling[10,11], in skeletal muscle of aged mice, and we generated mAktDKO mice, skeletal muscle-specific *Akt1/Akt2* double-knockout mice, by using the *Mlc1f*-Cre knock-in mice[28].

Mlc1f is expressed predominantly in fast-twitch muscle[34], with its promoter known to work little in satellite cells, undifferentiated precursors of myocytes[35]. Indeed, Akt activity was suppressed more efficiently in the fast-twitch muscle of the mAktDKO mice, and accordingly, the sarcopenic phenotypes of the mAktDKO mice were mainly observed in fast-twitch muscle, whose characteristics were found to coincide with those of sarcopenia in humans[27].

The reduced body weight of the mAktDKO mice was due to a reduction in skeletal muscle mass but not due to global growth retardation as was observed in the systemic *Irs1* knockout mice[36,37] or systemic *Akt1* knockout mice[24].

Fast-twitch muscle of the mAktDKO mice was shown to have lost not only muscle mass but also the characteristics of fast-twitch muscle, with type 2 myofibers decreased and enzymes involved in glycolysis down-regulated. Besides, the fast-twitch muscle of the mAktDKO mice showed dysfunction in mitochondria biogenesis, oxidative phosphorylation, and autophagy involved in the

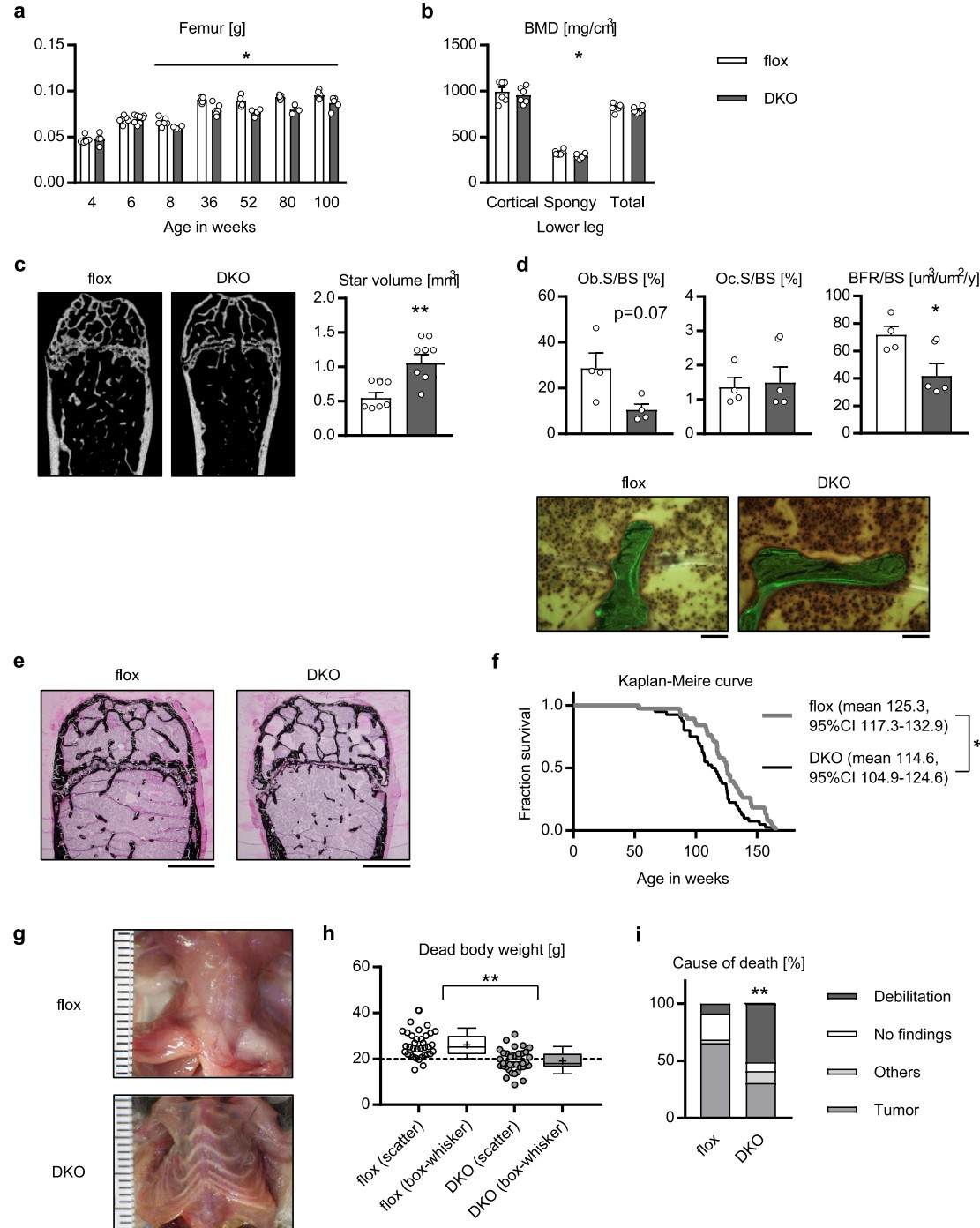

**Fig. 5 | Bone-associated and systemic phenotypes of the mAktDKO mice.**
**a** Femur weight of the mAktDKO mice at the age of the indicated weeks ($n$ = 3, 4, 5, 6, 7, or 8 mice per group, as shown by the number of data points on the graph, same as below). **b** Bone mineral density of lower legs at the age of 40 weeks analyzed by CT scanning ($n$ = 6 mice). **c**, Analysis of osteoporosis by micro-CT scanning of femur at the age of 52 weeks ($n$ = 6 mice). Scale bars: 500 µm. **d** Morphometric analysis of calcein-labeled femur at the age of 52 weeks ($n$ = 4 mice). Ob.S/BS: osteoblast surface/bone surface, Oc.S/BS: osteoclast surface/bone surface, BFR/BS: bone formation rate/bone surface. Scale bars: 50 µm. **e** von Kossa staining of femur at the age of

52 weeks. Scale bars: 500 µm. **f** Kaplan–Meire curve for survival ($n$ = 38 or 40 mice). **g** Macroscopic image of thorax of a dissected dead body. Scale graduation: 1 mm. **h**, **i** (**h**) Scatter plot (left) and box and whisker plot (right) of dead body weight, and (**i**) cause of death, of mice whose dead body was retrieved without severe deterioration ($n$ = 35 or 39 mice). Box: median and quartiles, whisker: 10% and 90% percentiles, +: mean. Values of the data are expressed as mean ± SEM except in **f**, **h**, **i**. *$P$ < 0.05, **$P$ < 0.01. Unpaired 2-tailed $t$-test (**a**–**d**, **h**), logrank test (**f**), and Chi-square test with Yates' correction (**i**) were used for assessment respectively, and the exact $P$ values are provided in Supplementary Data 2. Source data are provided as a Source Data file.

elimination of abnormal mitochondria, and thus systemic skeletal muscle, mainly composed of fast-twitch muscle, was considered incapable of producing enough ATP anyhow, explaining why the mAktDKO mice exhibited an impairment in exercise endurance, as well as a decline in grip strength. Grip strength decline and slower

gait speed are included in the criteria for the diagnosis of sarcopenia[38,39], both of which reflect impaired instant power. Our results also suggest that impaired exercise endurance could be another characteristic of sarcopenia, leading to the inability to retain posture, which could be clinically important in aged people.

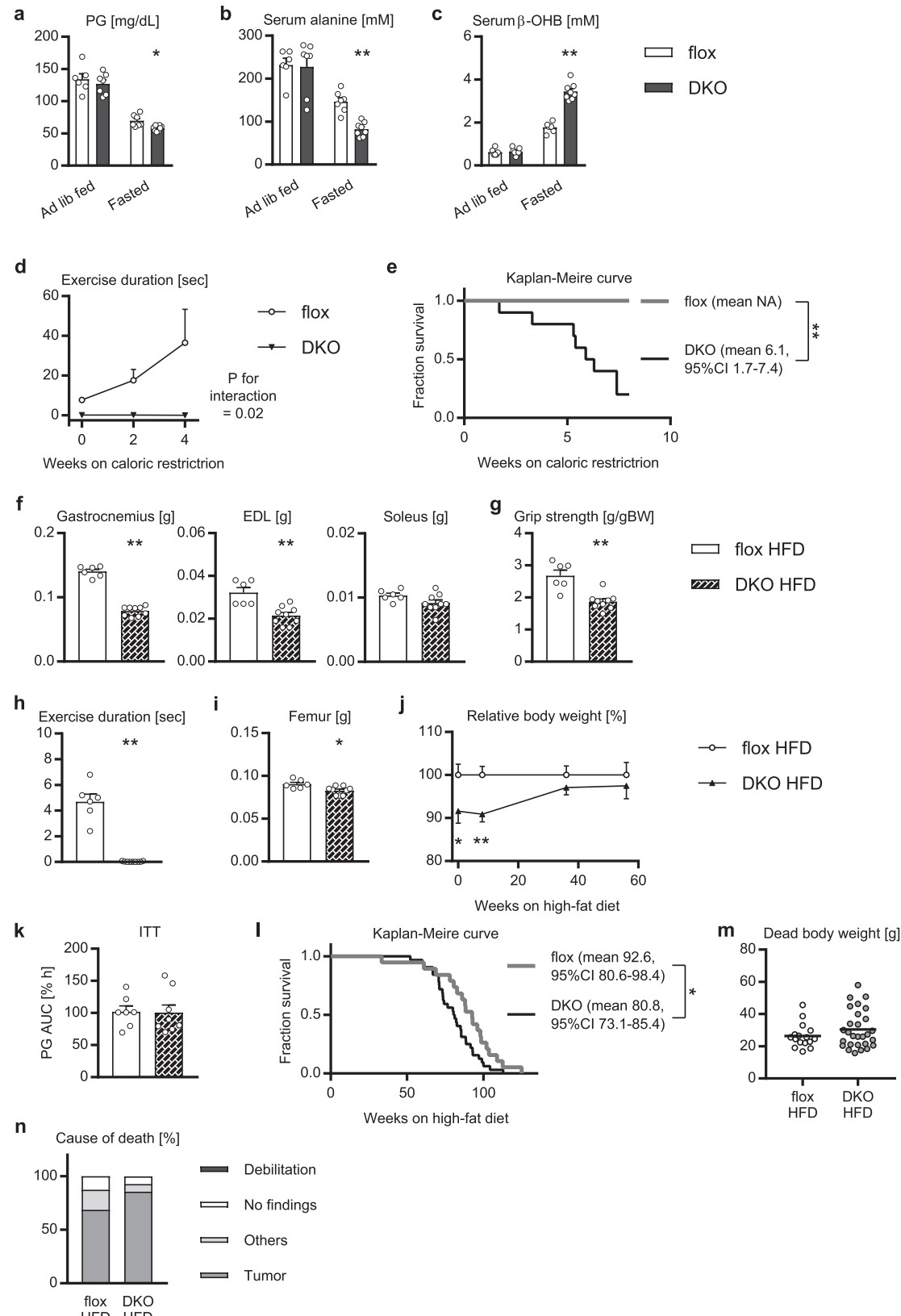

In addition to sarcopenia, the mAktDKO mice exhibited bone loss, which we call osteosarcopenia. Reduced mechanical loading is among its potential mechanisms, given that the mAktDKO mice exhibited decreased muscle mass and impaired motor function. However, it is known that cortical bones are mainly affected by mechanical loading[40], whereas spongy bones were mainly affected in the mAktDKO mice. Moreover, osteopenic changes were observed before muscle loss became evident. These data suggest that some myokines or metabolites derived from skeletal muscle regulated by Akt (and presumably through FoxOs) could affect osteogenesis, which should be explored and elucidated in future studies.

**Fig. 6 | Phenotypes of the mAktDKO mice in under- and over-nutrition. a** Plasma glucose (PG), **b** serum alanine levels, and **c** serum β-hydroxybutyrate (β-OHB) levels, in the mAktDKO mice at the age of 60 weeks in the *ad libitum* fed state and in the fasted state for 24 h (*n* = 6, 7, or 8 mice per group, as shown by the number of data points on the graph, same as below). **d** Exercise duration of the mAktDKO mice during caloric restriction for 4 weeks from the age of 90 weeks (*n* = 6 mice). **e** Kaplan–Meire curve for survival in the mAktDKO mice subjected to caloric restriction from the age of 90 weeks (*n* = 9 or 10 mice). (**f, i**) Tissue weight, **g** grip strength, and **h** exercise duration, of the mAktDKO mice fed with high-fat diet for 40 weeks (*n* = 6 or 9 mice). **j** Relative body weight of the mAktDKO mice fed with high-fat diet for the indicated weeks compared to the control mice (*n* = 11 or 17 mice). **k** Area under the curve (AUC) of relative PG in insulin tolerance test (ITT) after intraperitoneal injection of human regular insulin (2.5 U/kg BW) in the mAktDKO mice fed with high-fat diet for 40 weeks (*n* = 7 or 8 mice). **l** Kaplan–Meire curve for survival of the mAktDKO mice fed with high-fat diet (*n* = 19 or 32 mice). **m** Scatter plot of dead body weight, and (**n**) cause of death, of mice whose dead body was retrieved without severe deterioration (*n* = 16 or 28 mice). Values of the data are expressed as mean ± SEM except in (**e, l, m, n**). *$P < 0.05$, **$P < 0.01$. HFD: high-fat diet. Unpaired 2-tailed *t*-test (**a**–**c, f**–**k, m**), repeated measure analysis of variance (**d**), logrank test (**e, l**), and Chi–square test with Yates' correction (**n**) were used for assessment, respectively, and the exact *P* values are provided in Supplementary Data 2. Source data are provided as a Source Data file.

Most of these phenotypes associated with knocking out of *Akt1/2* in skeletal muscle appeared at the age of 8 weeks or much later, possibly due to some compensatory mechanisms that might deteriorate with aging, and at least mitochondrial dys-function was shown likely to be compensated by activation of the AMPK pathway in the younger mAktDKO mice. This was in contrast to the previous model mice designed to modulate other transducers of insulin/IGF-1 signaling thus exhibiting more radical phenotypes of growth retardation rather than accelerated aging, such as skeletal muscle-specific IR and IGF1R knockout mice exhibiting reduced body weight at the age of 3 weeks and dying before the age of 30 weeks[14].

Indeed, while the mAktDKO mice lived for over 100 weeks on average, their mean survival was still significantly shortened compared to that in the control mice. Moreover, about half of the mAktDKO mice died from debilitation, whereas more than half of the control mice died from tumor. It is noteworthy that knocking out of just a single kinase in just a single tissue not only mildly affects lifespan but even drastically affects the cause of death.

Autophagy in skeletal muscle is assumed to contribute to the maintenance of blood glucose by supplying alanine, serving as a substrate for gluconeogenesis in the liver. Indeed, lower blood glucose levels accompanied by lower serum alanine levels and elevated ketone bodies were observed in the fasted mAktDKO mice, as in tissue-specific AMPK knockout mice, a model of autophagy failure in skeletal muscle[41]. Caloric restriction resulted in further shortening of exercise duration and even lifespan in the mAktDKO mice, which argues against excessive diet therapy in sarcopenic patients and supports the importance of the patient-oriented approach recommended by the recent clinical guidelines[42,43].

In contrast, the sarcopenic phenotypes of the mAktDKO mice were not rescued with over-nutrition, or high-fat diet feeding. Although no synergistic aggravation of insulin resistance was observed in this potential model of sarcopenic obesity with the inter-genotype difference in systemic insulin resistance due to sarcopenia probably masked by that due to increased adiposity, they still exhibited reduced lifespan. Most of the mAktDKO mice fed with a high-fat diet died from tumor, just as the control mice did. It was also observed that proliferation of subcutaneously transplanted melanoma was accelerated in the mAktDKO mice, suggesting that skeletal muscle could play an important role in regulating the growth of tumor in remote tissues, possibly by some secretory factors, which could explain poor prognosis of cancer in patients with sarcopenic obesity.

Then we went on to identify the key pathway downstream of Akt to account for the phenotypes of the mAktDKO mice with a focus on the FoxO pathway regulating protein degradation and the mTOR pathway regulating protein synthesis. The mAkt/TscTKO mice, in which the latter pathway was activated, failed to restore the reduced skeletal muscle and bone mass observed in the mAktDKO mice, but completely rescued the weakened grip strength, which was attributed to recovery of expression of the genes involved in glycolysis. Mitochondria were almost doubled, albeit with an abnormal appearance, and exercise endurance was restored only partially. Despite recovered expression of the genes involved in oxidative phosphorylation and autophagy, *Sod2* was downregulated just as in the mAktDKO mice, suggesting that an increase in the number of mitochondria alone may not be sufficient for the maintenance of function without expanded anti-oxidant capacity in skeletal muscle. Besides, the mAkt/TscTKO mice exhibited growth retardation, accompanied by lower *Igf1* expression, just as skeletal muscle-specific *Tsc1* knockout mice did[44]. Although knocking out of *Tsc2* could also affect the mTOR-independent pathways, which might have affected the phenotypes of the mAkt/TscTKO mice, overall, activation of the mTOR pathway in skeletal muscle was not considered a good strategy for treating osteosarcopenia.

In contrast, additional knocking out of *Foxo1/4* successfully restored the reduced muscle mass seen in the mAktDKO mice, as in the knockout mice of FoxOs, which were protective against muscle loss[45]. Motor dys-function, bone loss, and reduced lifespan were all restored as well in the mAkt/FoxoQKO mice, with the only exception of systemic insulin resistance which exhibited only partial recovery, reflecting a mixed result of recovered skeletal muscle mass and still impaired insulin-dependent glucose uptake. Moreover, administration of a FoxO inhibitor, AS1842856, showed a partial but significant recovery of the reduced fast-twitch muscle mass of the mAktDKO mice. These data suggest that the FoxO pathway in skeletal muscle could be a good therapeutic target in this context, and it should be investigated in future works how to optimize the dosage and delivery of FoxO inhibitors, formally developed as anti-diabetic drugs[32], to prove their effectiveness against sarcopenia.

A comparison of the mAkt/TscTKO and mAkt/FoxoQKO mice suggests that the genes involved in glycolysis, oxidative phosphorylation, and autophagy could be regulated by both the FoxO pathway and mTOR pathway in a redundant manner. The FoxO pathway could positively regulate skeletal muscle mass, type 2 myofibers, and anti-oxidant capacity, whereas the mTOR pathway could be a major regulator of type 1 myofibers and quantity of mitochondria (Fig. 9a). Although various reports have tried to examine involvement of the FoxO pathway and the mTOR pathway in some of these changes in skeletal muscle[15–17,19,21,44,45], we believe that our data contribute to comprehensive understanding of Akt-centered mechanisms underlying sarcopenia, including different roles of the downstream pathways, with subsequent effects on bone and lifespan.

In lower organisms, lifespan is thought to be reduced by insulin/IGF-1 signaling with FoxOs as the key downstream molecules[1,2], and it was reported to be prolonged in flies with muscle-specific genetic modification in which insulin signaling was suppressed and FoxO was up-regulated[46,47]. These previous studies share the same key molecules with our mouse model, but are diametrically opposed to ours with regard to their effects on lifespan. Our data clearly show that, contrary to the established theory, suppressed insulin/IGF-1 signaling followed by activation of FoxOs reduces lifespan in skeletal muscle in mammals (Fig. 9b).

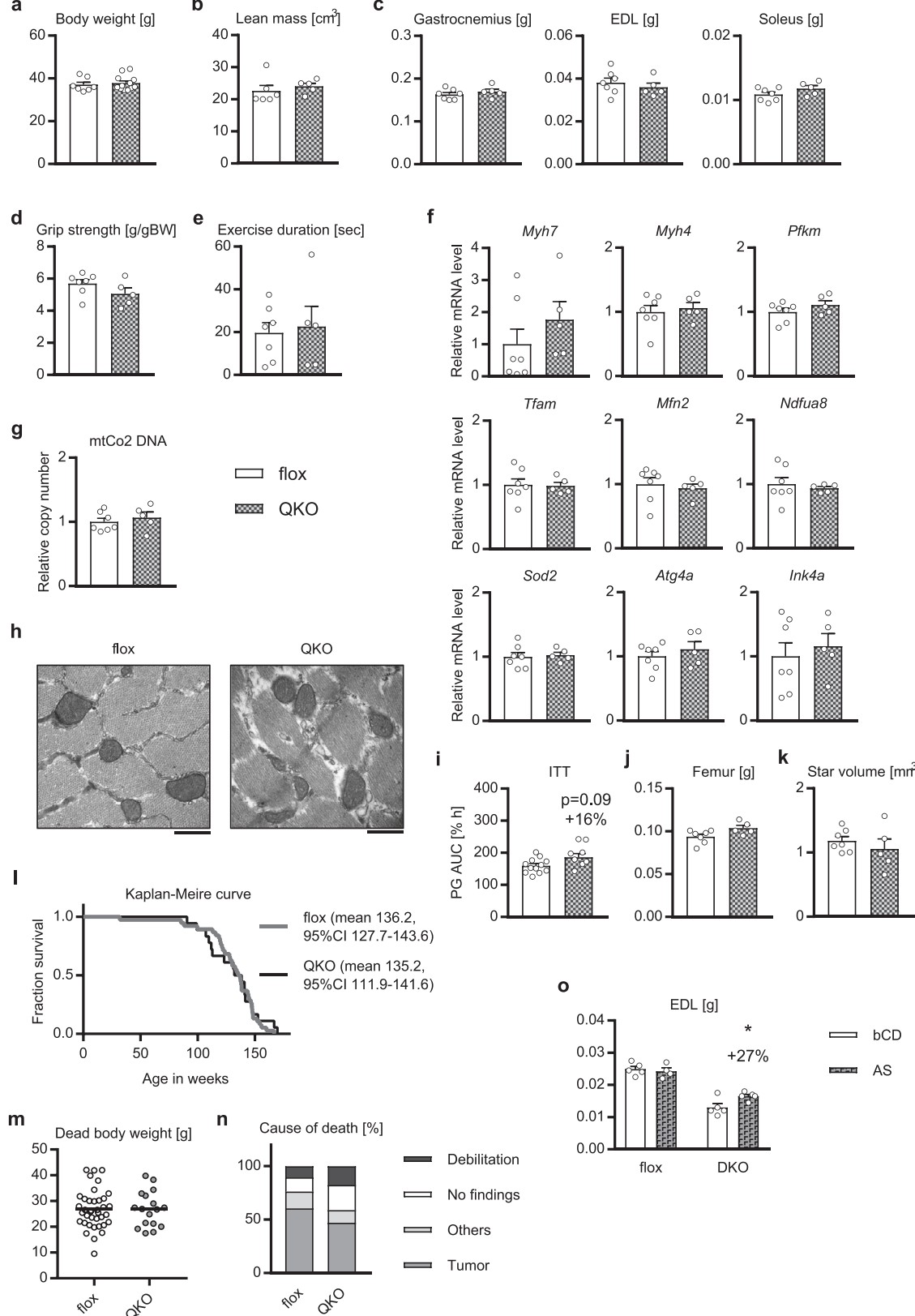

In conclusion, suppressed Akt activity in skeletal muscle results in accelerated aging, i.e., premature osteosarcopenia, systemic insulin resistance, and reduced lifespan. These changes may be amenable to nearly complete reversal by inactivation of FoxOs, which are thus thought to represent likely therapeutic targets in coping with skeletal muscle-centered aging.

## Methods

### Mice

All mice were housed under a 12-hour light/12-hour dark cycle at macroenvironmental temperature and humidity ranges of 20 to 22 °C and 40 to 60%, respectively, and had free access to sterile water and pellet food, Oriental MF diet (Oriental Yeast, consisting of 23.6% (v/v)

**Fig. 7 | Phenotypes of the mAkt/FoxoQKO mice. a–e, j** Body weight (*n* = 7 or 12 mice per group, as shown by the number of data points on the graph, same as below) (**a**), lean body mass analyzed by systemic CT scanning (*n* = 5 or 6 mice) (**b**), **c, j** tissue weight (*n* = 5 or 7 mice), **d** grip strength (*n* = 5 or 7 mice), and **e** exercise duration (*n* = 5 or 7 mice), of the mAkt/FoxoQKO mice at the age of 52 weeks. **f, g** Gene expression (**f**), and DNA copy number, of EDL at the age of 52 weeks analyzed by RT-PCR (*n* = 5 or 7 mice) (**g**). **h** Electron microscopic image of vertical sectioning of EDL at the age of 52 weeks. Scale bars: 500 nm. **i** Area under the curve (AUC) of relative plasma glucose (PG) in insulin tolerance test (ITT) after intraperitoneal injection of human regular insulin (1.0 U/kg BW) at the age of 40 weeks (*n* = 7 or 12 mice). **k** Analysis of osteoporosis by micro-CT scanning of femur at the age of 52 weeks (*n* = 5 or 7 mice). **l** Kaplan–Meire curve for survival (*n* = 18 or 38 mice). **m, n** Scatter plot of dead body weight (**m**), and cause of death, of mice whose dead body was retrieved without deterioration (*n* = 17 or 38 mice) (**n**). **o** Skeletal muscle weight of the mAktDKO mice at the age of 90 weeks treated with AS1842856 at the dose of 100 mg/kg of BW daily for 4 weeks (*n* = 4 or 5 mice). bCD: β-cyclodextrin, AS: AS1842856. Values of the data are expressed as mean ± SEM except in (**l**–**n**). Unpaired 2-tailed *t*-test (**a**–**g**, **i**–**k**, **m**, **o**), logrank test (**l**), and Chi-square test with Yates' correction (**n**) were used for assessment respectively, and the exact *P* values are provided in Supplementary Data 2. Source data are provided as a Source Data file.

protein, 5.3% fat, 54.4% nitrogen-free extract, 6.1% ash, 2.9% fiber, and 7.7% water), unless otherwise indicated. C57BL/6 J mice were purchased from Oriental Yeast, and BKS.Cg-+ *Lepr^db*/+ *Lepr^db*/Jcl (*db/db*) mice from CLEA Japan[26]. Tissue-specific *Akt1/2* knockout mice were generated following a previous report:[48] *Akt1*-floxed[49], and *Akt2*-floxed mice[23] were crossed with *Mlc1f*-Cre knock-in mice[28], and subsequently crossed with *Foxo1*-floxed[19] and *Foxo4*-floxed mice[50] or with *Tsc2*-floxed mice[51]. Male mice were subjected to experiments, unless otherwise indicated. Body weight and blood glucose levels were matched among groups in experiments allocating mice to different interventions. Experiments in an *ad libitum*-fed state were performed immediately after the beginning of the light cycle. For monitoring food intake, mice were kept in individual cages for 24 hours. In caloric restriction experiments, 60% of the monitored food intake was given every another day[52]. To generate a diet-induced obesity model, we fed mice with High Fat Diet 32 (CLEA Japan, consisting of 25.5% protein, 32.0% fat, 29.4% nitrogen-free extract, 4.0% ash, 2.9% fiber, and 6.2% water) from the age of 10 weeks. The animal care and experimental procedures were approved by the Animal Care Committee of the Graduate School of Medicine, the University of Tokyo. All relevant ethical guidelines were followed.

## Metabolic studies

Mice were anesthetized for analysis of body composition by dual-energy X-ray absorptiometry with a LUNAR PixiMus2 scanner (GE Healthcare)[53], and for analysis of body composition and bone mineral contents by CT scanning with Latheta LCT-200 (Hitachi). For insulin tolerance tests (ITTs), mice received intraperitoneal administration of human insulin (Humalin R; Eli Lilly) in an *ad libitum*-fed state. For intraperitoneal glucose tolerance tests (IPGTTs), mice received intraperitoneal administration of glucose after an overnight fast. Blood glucose levels were measured using Glutest Every (Sanwa Kagaku Kenkyusho) at the indicated time points. Blood ketone bodies were measured using FreeStyle Presicion Neo (Abbott). Plasma alanine levels (Biowest) and plasma-free fatty acid levels (Wako) were measured using the corresponding enzyme-based quantification kit. ATP contents were analyzed using ATPLite (Perkin Elmer), according to the manufacturer's instruction. The investigators were blinded to the genotype of mice or intervention to which mice were subjected when assessing metabolic phenotypes.

## Glucose uptake assay ex vivo

After excision of skeletal muscle, the tendon on each side were ligated using a silk surgical thread, and the muscle was set across a plastic holder. After pre-incubation in 2 mM pyruvate in Krebs-Ringer bicarbonate HEPES buffer with 0.2% bovine serum albumin (KRBH/BSA) for 30 minutes at 37 °C, the muscle was incubated with 1 mM 2-deoxyglucose (2-DG), 1 mM L-Glucose (LG), and 10 nM insulin in KRBH/BSA in a glass vial for 10 minutes at 30 °C, followed by incubation at 30 °C for another 10 min after addition of 2-DG[14 C] (final concentration: 0.5 μCi/mL, 1 mM) and LG[3H] (5 μCi/mL, 1 mM) (American Radiolabeled Chemicals). The samples and buffers were continuously gassed with 95% $O_2$-5% $CO_2$. After washing with chilled KRBH/BSA, the muscle was resolved in 5 M NaOH overnight at room temperature, followed by the addition of HCl, methanol, and ACSII (Amersham). The 14 C and 3H specific activities were counted using a liquid scintillation counter (Packard Instrument), and the non-specific uptake of LG was subtracted from the total uptake of 2-DG, following a previous report[54].

## Motor function studies

Grip strength was measured using a traction meter (BS-TM-RM; BrainScience idea) by pulling a mouse on the mesh plate backward with the tail, and the average of three-time measurements was adjusted by body weight. For assessment of exercise duration, mice were subjected to running on a treadmill (MK-680C; Muromachi Kikai) for 900 seconds at a speed of 15 m/minute, and the number of times in which the mouse was unable to avoid electrical shocks were counted[55]. 900 seconds were divided by the count of electrical shocks, yielding the mean exercise duration in seconds. In the exercise tolerance test, mice were subjected to running on the treadmill without electrical shocks at a speed of 15 m/minute with a gradient of 14 degrees for 900 seconds in the older mice, just as in assessment of exercise duration and for extended duration of 3600 seconds in the younger mice.

## Survival

To follow up survival, multiple mice were housed in a cage with free access to sterile water and food. Dead bodies were retrieved, whose dead body weight was measured, followed by dissection for identification of the cause of death. Those with apparent tumor, lymph node swelling, hemorrhagic ascites, or hemorrhagic pleural effusion were judged to have died from tumor, whereas those with other macroscopic findings, including splenomegaly, hemorrhage, and skin injury, were judged to have died from other causes. Those without any macroscopic findings and with dead body weight less than 20 g, equivalent to the 10 percentile of body weight of the control mice, were judged to have died from debilitation.

## Chemical treatment

Streptozotocin (SIGMA) was diluted in citrate sodium buffer, and administered intraperitoneally at the dose of 150 mg/kg of body weight (BW) twice with an interval of 4 days[26]. Mice were used for experiments 4 weeks after the treatments. Leupeptin (Wako) was diluted in phosphate-buffered saline, and administered intraperitoneally at the dose of 40 mg/kg of BW 4 h before sacrifice[56]. Calcein (Dojindo) was diluted in NaHCO3 solution, and administered subcutaneously at the dose of 8 mg/kg of BW twice, 8 days and 2 days before sacrifice[57]. AS1842856, or 5-amino-7-(cyclohexylamino)−1-ethyl-6-fluoro-4-oxo-1,4-dihydroquinoline-3-carboxylic acid (Universal Biologicals), was diluted in 6% β-cyclodextrin (Sigma), and administered orally at the dose of 100 mg/kg of BW[32] once daily for 4 weeks.

## Culture of calvarial cells

For isolation of neonatal murine calvarial cells, calvaria were dissected from neonatal mice and digested in a digestion solution, in which 1 mg/mL of collagenase type I (Wako) and 2 mg/mL of dispase

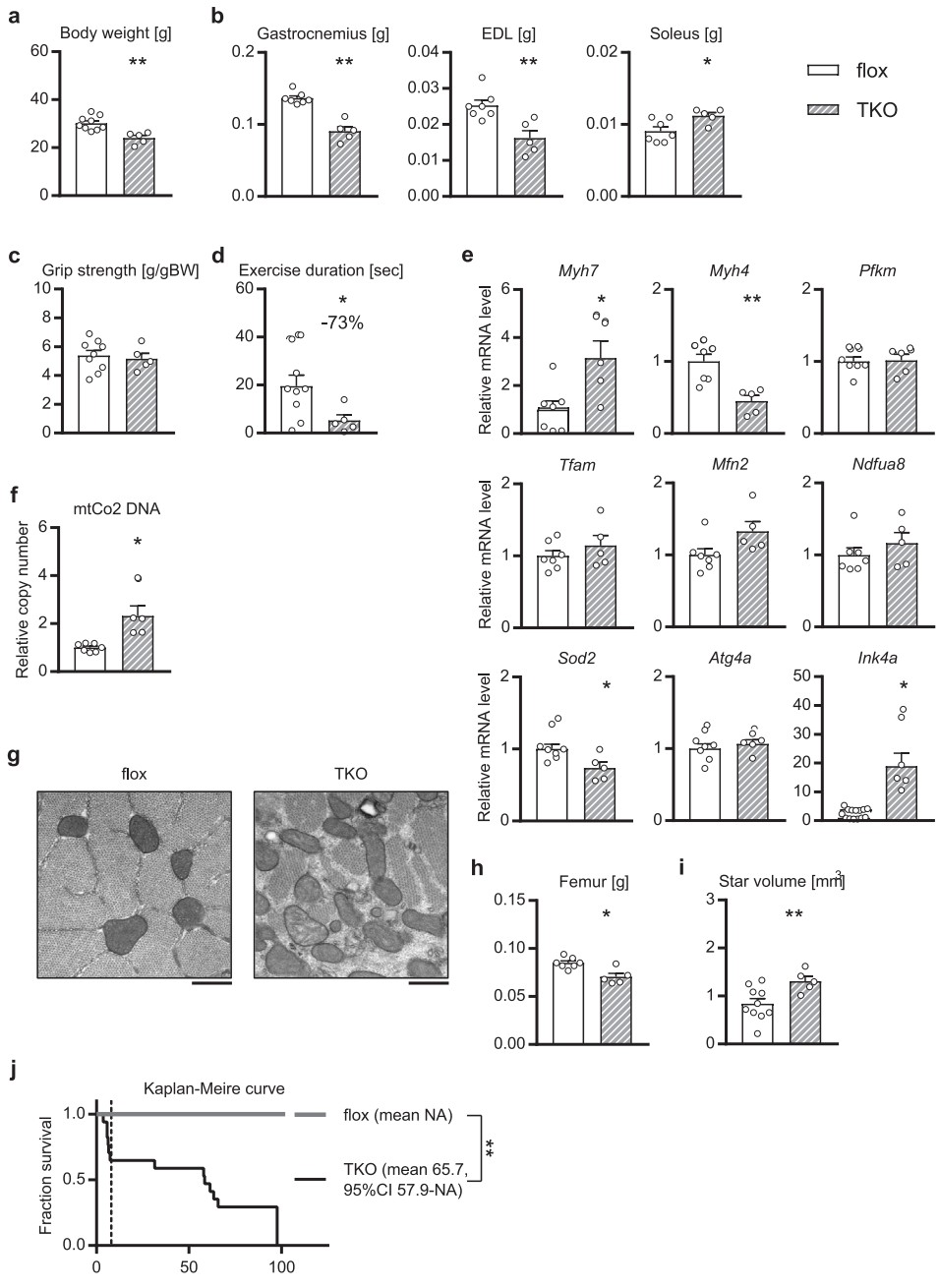

**Fig. 8 | Phenotypes of the mAkt/TscTKO mice. a** Body weight (*n* = 5 or 9 mice per group, as shown by the number of data points on the graph, same as below), **b**, **h** tissue weight (*n* = 5 or 7 mice), **c** grip strength (*n* = 5 or 9 mice), and **d** exercise duration (*n* = 5 or 9 mice), of the mAkt/TscTKO mice at the age of 52 weeks. **e** Gene expression, and **f** DNA copy number, of EDL at the age of 52 weeks analyzed by RT-PCR (*n* = 5 or 7 mice). **g** Electron microscopic image of vertical sectioning of EDL at the age of 60 weeks. Scale bars: 500 nm. **i** Analysis of osteoporosis by micro-CT scanning of femur at the age of 52 weeks (*n* = 5 or 10 mice). **j** Kaplan–Meire curve for survival (*n* = 15 or 17 mice) with the mean survival of 8-week survivors (*n* = 11 or 15 mice). Values of the data are expressed as mean ± SEM except in **j**. Unpaired 2-tailed *t*-test (**a**–**f**, **h**, **i**) and logrank test (**j**) were used for assessment, and the exact *P* values are provided in Supplementary Data 2. \**P* < 0.05, \*\**P* < 0.01. Source data are provided as a Source Data file.

II (Wako) were dissolved in α-MEM (Gibco). Isolated cells were cultured in α-MEM containing 10% fetal bovine serum and seeded in a culture dish (10 cm dish for one mouse). After incubation for 2 days, cells were resuspended in an osteogenic medium (50 mg/mL ascorbic acid, 10 nM dexamethasone and 10 mM β-glycerophosphate) and seeded in 24-well plates (1.0 × 10⁴ cells per well). The culture medium was changed every third day. For ALP staining and Alizarin Red S staining, cultured cells were fixed with 4% paraformaldehyde (PFA) for 15 min on ice, and then stained for 15 min in room temperature with ALP staining solution (Napthol AS-MX

phosphatase, 0.06 mg/mL; N,N-dimethylformamide, 1%; and Fast blue BB salt, 1 mg/mL, in 0.1 M Tris-HCl, pH 8.0), or with Alizarin Red S solution (0.02 g/mL, pH 4.2). In either staining, the staining solution was washed away with tap water, and stained cells were air dried, followed by microscopic observation[58].

### Culture of melanoma cells and subcutaneous transplantation
B16F1 cells (ATCC CRL-6323), a melanoma cell line, were cultured in DMEM (Sigma) supplemented with 10% FBS (Biowest), without authentication thereafter. Expanded cells were resuspended in

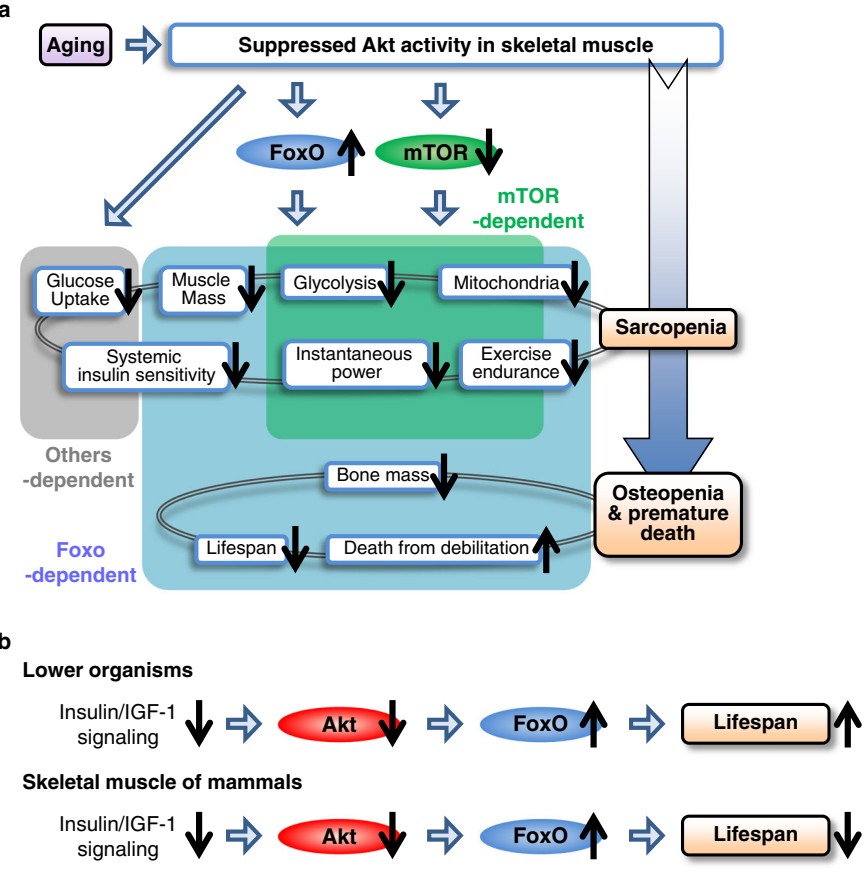

**Fig. 9 | Schematic summary. a** Role of Akt in skeletal muscle in protection against sarcopenia, osteopenia, and reduced lifespan in mice. **b** Effects of impaired insulin/IGF-1 signaling on lifespan in lower organisms and in skeletal muscle of mammals.

phosphate-buffered saline, and 0.2 mL of the suspension, containing $1 \times 10^6$ cells, was injected into the back of mice subcutaneously[59]. The greatest diameter of tumor was measured with an analog caliper (Niigata Seiki).

## Gene expression analysis

RNeasy Mini Kit was used to prepare total RNA from mouse tissues, using homogenates of frozen tissues in RLT Buffer after incubation with Proteinase K (both from Qiagen). Reverse-transcription reaction was carried out with a High Capacity cDNA Reverse Transcription Kit (Applied Biosystems), after treatment with DNase (Promega). DNeasy Mini Kit (Qiagen) was used to prepare genomic and mitochondrial DNA from mouse tissues. Quantitative PCR analyzes were performed using ABI Prism 7900, with TaqMan Gene Expression Master Mix or Power SYBR Green PCR Master Mix (Applied Biosystems)[26]. The relative mRNA expression levels were normalized by the expression level of *Ppia3* (encoding Cyclophilin A), and the relative mitochondrial DNA copy number was normalized by the copy number of *Mapk1*. TaqMan probes were synthesized by Applied Biosystems: *Akt2*, forward TCGA-GAGGACCTTCCATGTAGAC, reverse TGGATAGCCCGCATCCA, probe TCCAGATGAGAGGGAAGA; *Ppia3*, forward GGTCCTGGCATCTTGTC-CAT, reverse CAGTCTTGGCAGTGCAGATAAAA, probe CTGGACCAAA-CACAAACGGTTCCCA. Inventoried TaqMan assays were synthesized by Applied Biosystems; *Adipoq*, Mm00456425_m1; *Akt1*, Mm00437443_m1; *Arg1*, Mm00475988_m1; *Atg12*, Mm00503201_m1; *Cat*, Mm00437992_m1; *Ccl2*, Mm00441242_m1; *Cpt1b*, Mm00487200_m1; *Emr1*, Mm00802529_m1; *Fgf21*, Mm00840165_g1; *Fis1*, Mm00481580_m1; *Foxo3*, Mm01185722_m1; *Foxo4*, Mm00840140_g1; *Hk2*, Mm00443385_m1; *Lep*, Mm00434759_m1; *Mfn2*, Mm00500120_m1; *Nppa*, Mm01255747_g1; *Nppb*, Mm01255770_g1; *Nrf1*, Mm01135606_m1; *Opa1*, Mm00453879_

m1; *Pfkm*, Mm00445461_m1; *Ppara*, Mm00440939_m1; *Ppargc1a*, Mm01208835_m1; *Sod2*, Mm01313000_m1; *Slc2a4*, Mm01245502_m1; *Tfam*, Mm00447485_m1; *Tnf*, Mm99999068_m1; *Ucp1*, Mm01244861_m1.

## Primers

Primers for quantitative PCR analyzes were adopted from PrimerBank (http://pga.mgh.harvard.edu/primerbank/), unless otherwise described, except *Cox8b* and *Foxo1*, which were designed in Primer3 (http://bioinfo.ut.ee/primer3-0.4.0/). Oligonucleotides were synthesized by Invitrogen.

| Primers for RT-PCR (mRNA expression level) | | |
|---|---|---|
| *Atg4a*[60] | Forward | CCCTCACACAACCCAGACTT |
| | Reverse | CCCCTGTGGTTGTCACTTCT |
| *Cox8b* | Forward | TGCGAAGTTCACAGTGGTTC |
| | Reverse | TGCTGCGGAGCTCTTTTTAT |
| *Fbxo32* | Forward | CAGCTTCGTGAGCGACCTC |
| | Reverse | GGCAGTCGAGAAGTCCAGTC |
| *Foxo1* | Forward | AAGAGCGTGCCCTACTTCAA |
| | Reverse | CAGGTCATCCTGCTCTGTCA |
| *Gdf11* | Forward | CTACCACCGAGACGGTCATAA |
| | Reverse | CCGAAGGTACACCCACAGTT |
| *Igf1* (circulating form) | Forward | GTGGACCGAGGGGCTTTTACTTC[61] |
| | Reverse | ACATTCTGTAGGTCTTGTTTCC[62] |
| *Igf1* (local form) | Forward | = *Igf1* (circulating form) |
| | Reverse | CGATAGGGACGGGGACTTC[62] |

| Ink4a (Cdkn2a)[63] | Forward | CGTACCCCGATTCAGGTGAT |
|---|---|---|
| | Reverse | TTGAGCAGAAGAGCTGCTACGT |
| Map1lc3b[64] | Forward | CGGAGCTTTGAACAAAGAGTG |
| | Reverse | TCTCTCACTCTCGTACACTTC |
| Myh1 | Forward | CTCTTCCCGCTTTGGTAAGTT |
| | Reverse | CAGGAGCATTTCGATTAGATCCG |
| Myh2 | Forward | TAAACGCAAGTGCCATTCCTG |
| | Reverse | GGGTCCGGGTAATAAGCTGG |
| Myh4 | Forward | CCGCATCTGTAGGAAGGGG |
| | Reverse | GTGACCGAATTTGTACTGAGTGT |
| Myh7 | Forward | GCTACGCTTCCTGGATGATCT |
| | Reverse | CCTCTTAGTGTTGACAGTCTTCC |
| Ndufa8 | Forward | GGAGGAGGTGAAAGTCAGCTC |
| | Reverse | GCATGTTGGAGTAATCAAGGCAA |
| Runx2 | Forward | AGAGTCAGATTACAGATCCCAGG |
| | Reverse | TGGCTCTTCTTACTGAGAGAGG |
| Sirt1 | Forward | CAGCCGTCTCTGTGTCACAAA |
| | Reverse | GCACCGAGGAACTACCTGAT |
| Sirt3 | Forward | ATCCCGGACTTCAGATCCCC |
| | Reverse | CAACATGAAAAAGGGCTTGGG |
| Trim63 | Forward | CCAGGCTGCGAATCCCTAC |
| | Reverse | ATTTTCTCGTCTTCGTGTTCCTT |
| Tsc2 | Forward | GAGCTGATTAACTCGGTGGTC |
| | Reverse | GGCCAGGTCCCTTTCTTCC |
| Primers for RT-PCR (DNA copy number)[65] | | |
| Mapk1 | Forward | GCTTATGATAATCTCAACAAAGTTCG |
| | Reverse | ATGTTCTCATGTCTGAAGCG |
| mtCo2 | Forward | GCCGACTAAATCAAGCAACA |
| | Reverse | CAATGGGCATAAAGCTATGG |

## RNA sequencing

Approximately 1 μg total RNA was subjected to rRNA removal using NEBNext rRNA Depletion Kit (NEB) and then construction of libraries using NEBNext Ultra II Directional RNA Library Prep Kit for Illumina (NEB). The generated libraries were sequenced by NextSeq500 (Illumina) with paired-end, 37 bp reads. Basecall and demultiplexing were performed with bcl2fastq (v2.19.0, Illumina). The resulting raw reads were mapped to the mouse genome (mm10) using STAR (v2.6.2b)[66], with default parameters and expressed genes were counted using RSEM (v1.3.1)[67]. Differential Expression analysis based on negative binomial distribution using regression models of normalized count data and $p$ value adjustment by the Benjamini–Hochberg method were performed using DESeq2 (v1.30.0)[68], and adjusted $p$ value < 0.01 and $q$ value < 0.05 were considered as significantly different.

Genes annotated with mitochondria, ubiquitin and autophagy were enriched using AmiGO database (http://amigo.geneontology.org/amigo). Gene set enrichment analysis was performed using GSEA v4.1.0[69].

## Antibodies

Anti-dystrophin (ab15277, 1:100 for immunostaining) and anti-VDAC1 (ab15895, 1:1000 for western blotting hereinafter) antibodies were purchased from abcam; anti-pY (4G10, 1:1000) antibody was from Upstate; and the others, anti-p-Akt (9271, 1:1000), anti-Akt (9272, 1:1000), anti-p-AMPKα (2531, 1:1000), anti-AMPKα (2532, 1:1000), anti-p-AS160 (4288, 1:1000), anti-AS160 (2447, 1:1000), anti-p-FoxO1 (9461, 1:1000), anti-FoxO1 (9454, 1:1000), anti-FoxO4 (9472, 1:1000), anti-GAPDH (2118, 1:5000), anti-LC3B (2775, 1:1000), anti-p-mTOR (2971, 1:1000), anti-mTOR (2972, 1:1000), antibodies were from Cell Signaling. Secondary antibody for immunostaining (Alexa 594 Goat

Anti-Rabbit; A11037, 1:400) was purchased from Invitrogen, and those for western blotting (HRP-linked Anti-Mouse [from Sheep] and Anti-Rabbit [from donkey]; NA931 and NA934 respectively, both 1:2000) were from GE Healthcare Life Sciences.

## Western blot analysis

Tissues were quickly excised and frozen in liquid nitrogen. For assessment of insulin signaling, mice were anesthetized and administered with 5 units of Humalin R from inferior vena cava. To prepare total lysates, frozen tissues were homogenized in the liver buffer, containing 25 mM Tris-HCl at pH 7.4, 10 mM sodium orthovanadate, 10 mM sodium pyrophosphate, 100 mM sodium fluoride, 10 mM EDTA, 10 mM EGTA, and COMPLETE (Roche), and subjected to ultracentrifugation using Optima LE-80K Ultracentrifuge (Beckman Coulter) and the 70.1Ti rotor (280000 g for 1 h). To prepare the mitochondrial fractions, fresh tissues were homogenized in 0.25 M sucrose buffer, containing 10 mM Tris-HCl at pH 7.4 and 1 mM EDTA, followed by centrifugation at 900 g for 10 minutes. The supernatant was subjected to centrifugation at 6000 g for 10 minutes, and the precipitates were resuspended in the liver buffer[26,70]. The samples were resolved on SDS-PAGE and transferred to Hybond-P membranes (GE Healthcare)[26]. The membranes were incubated with primary antibodies, after blocking with bovine serum albumin (Sigma) resolved in Tris Buffered Saline (TBS) with Triton X-100 (anti-pY, and anti-p-mTOR antibodies), blocking solution for DIG (Roche) diluted in TBS with Tween 20 (WAKO) (anti-p-Foxo1, anti-p-AS160, and anti-AS160 antibody), or skim milk (Wako) resolved in TBS with Triton X-100 (the other antibodies). Bound primary antibodies were detected with HRP-conjugated secondary antibodies, using ECL detection reagents (GE Healthcare). Exposed films (Fujifilm) were processed by OMAT10000 (Kodak). Uncropped and unprocessed images are shown in Supplementary Fig. 11.

## Histological analysis of skeletal muscle

For immunostaining, tissues were fixed with 4% PFA and embedded in paraffin. Sections were incubated with the primary antibody after blocking with goat serum resolved in TBS in Tween 20, followed by incubation with the secondary antibody. Immunostained slides were observed using a fluorescence microscope BZ-X700 (Keyence) and cross-sectional areas of myofibers were analyzed by the BZ-H3C/Hybrid Cell Count Module (Keyence). For ATPase staining, tissues were rapidly frozen in liquid nitrogen-cooled isopentane, and transverse serial sections were incubated with barbital acetate solution at pH 4.11 or 10.43, followed by incubation with ATP incubating solution. For senescence-associated β-galactosidase staining, tissues were fixed in 4% PFA, followed by staining using the corresponding kit (K320-250; BioVision) according to the manufacturer's instructions. Macroscopic images were acquired with Optio WG-2 (RICOH)[71].

## Histological analysis of femurs

Femurs were fixed in 70% ethanol overnight, and the fixed distal left femurs were subjected to micro-CT scanning using a ScanXmate-L090 Scanner (Comscantechno) at 10.351 μm slices. The longitudinal images were analyzed using TRI/3D-BON software (RATOC Systems) for the calculation of structural indices. The fixed right femurs were embedded in glycol methacrylate after undecalcification, and 3 μm sections in the distal femur were longitudinally cut using a microtome (RM2255; Leica), followed by toluidine blue staining or von Kossa staining. The toluidine blue-stained sections were subjected to morphometric analyzes using a microscope (BX51; Olympus) and OsteoMeasure software (Osteometrics)[57].

## Transmission electron microscopic analysis

Tissues were prefixed with 2% PFA and 2% glutaraldehyde in 30 mM HEPES buffer (pH 7.4) overnight at 4 °C, followed by post-fixation with

an aldehyde OsO4 mixture (1.25% GA, 1% PFA, 0.32% K3[Fe(CN)6], and 1% OsO4 in 30 mM 9 HEPES buffer (pH 7.4)). Fixed samples were subjected to dehydration in ethanol series and then infiltrated with propylene oxide and embedded in Quetol 812 (Nisshin EM Corporation). Resin blocks were sectioned at 80 nm thicknesses with an ultramicrotome (Leica EM UC7; Leica), contrasted with uranyl acetate and lead citrate, and observed under a transmission electron microscope (JEM-1400; JEOL Ltd.)[72]. The percentage of mitophagy-like events was calculated following a previous report[73].

## Statistics

Values of the data are expressed as mean ± SEM and statistical significance was set at $P < 0.05$, unless otherwise indicated. Statistical analyzes were performed using unpaired 2-tailed $t$-test, chi-square test with Yates' correction, repeated measure analysis of variance, or one-way ANOVA (ANalysis Of VAriance) with post-hoc Tukey's Honest Significant Difference (HSD). The survival rate of mice was summarized as a cumulative proportion with the Kaplan-Meier method and compared by use of the log-rank test. These analyzes were performed with EZ-R (version 4.0.3)[74], and graphs were drawn with GraphPad Prism (version 7.02).

## Data availability

Data of RNA sequencing are deposited in a public data depository with the accession number provided in the manuscript (GEO accession: GSE199074). The authors declare that all other data supporting the findings of this study are available within the manuscript and its Supplementary Information files or are available from the authors upon reasonable request. Source data are provided with this paper.

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

## Acknowledgements

We thank Ritsuko Hoshino, Fumiya Takahashi, Yuko Kanto, Yasuko Sakuma, Reiko Homma, Yuko Masaki, Asako Yoshida, Yoshiko Ito, Mizuki Chosa, Yasuko Ota, Ayumi Ohuchi, and Yumiko Kishida for

their excellent technical assistance and assistance with the animal care. We thank Prof. Steven J Burden (New York University), Prof. Morris J Birnbaum (Pfizer Worldwide Research), and Prof. Ronald A DePinho (Harvard Medical School) for providing us with genetically modified mice; Prof. Sakae Tanaka, Dr. Naoto Tokuyama, and Dr. Yuki Taniguchi (The University of Tokyo) for giving us advice on bone morphometric analysis; Prof. Hiroshi Takayangi, Dr. Kazuo Okamoto, and Dr. Takehito Ono (The University of Tokyo) for giving us advice on isolation and differentiation of calvarial cells. This work was supported by a grant for Translational Systems Biology and Medicine Initiative (TSBMI), Creation of Innovation Centers for Advanced Interdisciplinary Research Areas Program of the Ministry of Education, Culture, Sports, Science and Technology of Japan (MEXT); a Grant-in-Aid for Scientific Research (S) (15H05789, to K.U.) and a Grant-in-Aid for Scientific Research (C) (20K08857, to T.S.) by the MEXT; a grant from the Tokyo Society of Medical Sciences (to T.S.); and grants and endowments from Daiichi Sankyo Co., Ltd. (to K.U. and T.S.) and Kowa Co., Ltd. (to T.S.).

## Author contributions

T.S. developed the hypothesis, designed and performed the experiments, analyzed the data, and wrote the manuscript. T.U., K.S., and K.K. developed the hypothesis, designed and performed the experiments, and analyzed the data. M.S. and N.K. performed the experiments. Y.O. and H.A. designed the experiments. M.T-N. performed transmission electron microscopic analysis. T.C. performed transcriptome analysis. D.A., C.R.K., and T.N. generated genetically modified mouse models. T.Y. developed the hypothesis and designed the experiments. T.K. and K.U. developed the hypothesis, designed the experiments, analyzed the data, and reviewed and edited the manuscript.

## Competing interests

The authors declare no competing interests.
