## [Peer Review File · Nature Communications]

REVIEWER COMMENTS

Reviewer #1 (Remarks to the Author):

The aim of this manuscript is to characterize Akt1/2 dko mice and how their phenotype is in line with the phenotype of sarcopenic muscles. The rationale is that reduced Insulin/IGF1 signaling is detrimental for the muscle phenotype during aging. I recognize the enormous amount of work done in generating various, quite complex, muscle-specific transgenic lines. I do however find it difficult to follow the rationale sometimes. It is shown in many ways that reducing mTOR signaling, also a mediator of insulin signaling, either by rapamycin or genetic approaches can actually increase lifespan and improve the muscle phenotype in aging. What the authors show in this manuscript, in my opinion, is the effect of loss of Akt1/2 in muscle, while not addressing the aging issue directly. It is also too simplistic to consider Akt as the main downstream mediator of IGF1/insulin signaling. It is clearly an important one, but there are numerous other intracellular mediators of this pathway which do not go through Akt.

Specific points:

- Fig 1C should be quantified. The mentioned reduction in insulin-stimulated signaling is not that convincing. Is it significant? If so for which targets?
- Fig 3. The study of autophagy is not sufficient. They do not show flux measurements and the LC3 blot is of poor quality. They also mention mitophagy, but never show any on this.

Fig 3B shows a lot of holes in the fibers of the ko. Are these artefacts or are these really present in the fibers?

- The lack of induction of atrogen1 and murf1 is somewhat surprising and should be explained. They could increase the number of FoxO3 genes they control for. This is also not sufficient to conclude that protein degradation is not involved, particularly as they show an increase in p62 and LC3.
- Fig 6 would suggest that you can take out akt1, akt2, foxo1 and foxo4 from your muscles and all the rest seems normal. The suggested inhibition of foxo in sarcopenia seems a bit extreme, while they show there are no changes in numerous E3 ligases, most other groups find a close correlation between foxo activity and E3 ligase induction. This discrepancy needs to be explained.

Reviewer #2 (Remarks to the Author):

Key results

The goal of the present study was to determine whether disruption of the insulin/IGF-1 signaling by suppressing Akt activity in mouse skeletal muscle could accelerate sarcopenia and consequent systemic senescence by regulating anabolic and catabolic factors. The authors used Cre-LoxP system to generate skeletal muscle-specific Akt knockout mouse model and reported promoted sarcopenia and declines in grip strength and exercise endurance, accelerated osteopenia via impaired osteogenesis, induced insulin resistance, and premature death from debilitation. The underlying mechanisms were investigated by monitoring mitochondria-related parameters, and by genetically inactivating or activating the FoxO or mTOR pathway respectively. Their analyses reveal reduction in ATP content and abnormalities of mitochondria. In addition, data suggest that ablation of FoxOs can prevent sarcopenia, osteopenia, and premature death caused by Akt knockout, whereas activation of mTOR pathway has no major effect.

Validity

In general, the experimental designs, data collection, data analysis, and data interpretation were appropriate.

Significance

The overall scientific question is an important one and the results help to advance our

understanding of how Akt activity in skeletal muscle impact muscle physiology and systemic senescence.

Data and methodology

Results

In general, the quality of presentation can be improved by rearranging some of the figures and adjusting the size of some images.

- Specific comments and concerns:

1. For the first subsection, in line 114-117, the authors made the statement that phosphorylation of these factors was suppressed to a lesser extent in slow-twitch muscle. Please quantify the signals to support the statement. Also, if you are looking at the differences in phosphorylation of a protein, e.g. AKT, it would be good to also show the expression of total Akt in order to distinguish between a change in AKT expression vs a change in the phosphorylation status of AKT.
2. For the second subsection, in line 148-153, the authors made the comparison between mAktDKO and Akt1/Akt2KO. However, Fig. S2i-l only showed data from Akt1/Akt2KO at the age of 32 weeks. According to Fig. 1d and Fig. 2f, both body weight and Ad libitum PG before insulin challenge had only little or even no difference from AktDKO mice to the control mice at the age of 40 weeks, thus, data from Fig. S2i-l (32 weeks) may not be supportive enough to make that statement.
3. For the third subsection, in line 164-165 and line 188-189, it will guide the readers better if you can list out the genes.
4. For the fourth subsection, I have difficulty following the shift in focus. It will be better to explain how data in skeletal muscle promoted you to study senescence of other tissues.
5. Fig. 1c, as stated above, it would be better to have quantification of the signals and western blots on total proteins. In Fig1c and supplemental Fig2c, the substrate protein levels should also be assayed to insure the significant changes of phosphorylation.
6. Fig. 2c and 2e, labels for the x-axis or y-axis respectively would help the readers to understand the figure.
7. Fig. 3f, bands for LC3BI and LC3BII are not apparent or clear on this blot. Also, please label the positions clearly.
8. Fig. 4, please use the correct format of the units. For example, please use mm³ instead of mm3.
9. In supple Fig7b, the substrate protein levels should be provided.
10. Fig. S2c, blot for p-S6K is missing.
11. Akt(Thr308) is the positive upstream regulator of mTORC1. Can the authors explain the description in line 303-304? Many descriptions of Akt signaling pathway are inaccurate.
12. TSC1/2 can regulate biological processes through mTOR-dependent or -independent pathways, the potential role of mTOR-independent regulation should be discussed.

Methods

1. In exercise tolerance test, please provide reference(s) for the chosen parameters. Also, please explain why 15 min was chosen to measure the ATP content.
2. For "survival" section, please provide reference for the use of 20 g in the judgement of debilitation.
3. For "gene expression analysis", please provide company information for RNeasy Mini Kit.
4. For "primers" section, why are the primers sequences in both lowercase and uppercase?

Analytical approach

In general, the statistical methods are valid and correctly applied. With that said, it would be more appropriate to use Chi-square instead of t-test in the analysis of CSA distribution.

Clarity and context

Overall, the manuscript would benefit for a thorough review for grammar, misspelled words, and odd sentence structure.

References

- 1. In line 89 and 90, please add reference(s) for streptozotocin treatment and db/db mice.**
- 2. In line 282 and 283, please add reference(s) for the statements on FoxO and mTOR.**
- 3. In line 304-305, please add reference(s) for the statement on TSC2.**

Reviewer #3 (Remarks to the Author):

Sasako et al. present an interesting manuscript investigating the role of Akt in skeletal muscle to protect against sarcopenia and senescence. The present interesting evidence indicating that suppression of Akt1/2 in skeletal muscle associated with insulin resistance and aging could accelerate sarco-osteopenia and systemic senescence.

Hypothesis is that sarcopenia could be promoted by insulin resistance in skeletal muscle. This is interesting, but the more translatable question would be does increasing insulin sensitivity negate the sarcopenic phenotype?

Did the authors look at the effects of exercise in the muscle of the older mAktDKO mice? After the acute exercise bout – would be of interest to determine if other signaling pathways (i.e. Ampk / pAmpk) are similarly altered with age or still able to be stimulated with exercise, even in the absence of Akt.

For Fig. 4, why is the n only 3? An n=5-6 would provide greater support to the conclusions.

The authors mentioned that these experiments were performed in male and female mice, but it is not clear throughout if the data was pooled or is presented in male vs. female mice.

Responses to the Comments Made by the Reviewers

Responses to the comment made by the Reviewer 1:

We are extremely grateful to the Reviewer 1 for the very careful review of our manuscript and for the suggestions that made our manuscript markedly improved.

> Reviewer #1 (Remarks to the Author):

> The aim of this manuscript is to characterize Akt1/2 dko mice and how their phenotype is in line with the phenotype of sarcopenic muscles. The rationale is that reduced Insulin/IGF1 signaling is detrimental for the muscle phenotype during aging. I recognize the enormous amount of work done in generating various, quite complex, muscle-specific transgenic lines. I do however find it difficult to follow the rationale sometimes. It is shown in many ways that reducing mTOR signaling, also a mediator of insulin signaling, either by rapamycin or genetic approaches can actually increase lifespan and improve the muscle phenotype in aging. What the authors show in this manuscript, in my opinion, is the effect of loss of Akt1/2 in muscle, while not addressing the aging issue directly. It is also too simplistic to consider Akt as the main downstream mediator of IGF1/insulin signaling. It is clearly an important one, but there are numerous other intracellular mediators of this pathway which do not go through Akt.

As was discussed in the manuscript, skeletal muscle-specific IR and IGF1R knockout mice exhibited reduced body weight but died before the age of 30 weeks, and failed to serve as a model of sarcopenia. Among the downstream molecules, Akt is surely one of the major ones (Ref. #9), and it was natural to focus on the kinase and establish a knockout model. Skeletal muscle-specific knocking out of *Akt1* and *Akt2* might have been too simplistic, but we believe that it is because we selected the strategy that the importance of Akt (and the FoxO pathway) in skeletal muscle was clearly highlighted.

As the Reviewer 1 pointed out, the mTOR pathway is surely an important player in the field of senescence (Ref. #28, newly added in the revised manuscript). To assess whether a reduction of the mTOR pathway may compromise aging phenotypes in the mAktDKO mice, we investigated the mAkt/TscTKO mice. Indeed, these mice showed rather reduced lifespan compared to the mAktDKO mice, although they still exhibited sarcopenia. It was intriguing that activation of the mTOR pathway in skeletal muscle could lead to increased mitochondria without expanded anti-oxidant capacity and

down-regulation of hepatic IGF-1 expression, which should be taken into account when considering activation of the pathway in humans. Moreover, we followed up survival of the mAkt/TscTKO mice, which turned out to be significantly shorter, further supporting our claim that activation of the pathway by knocking out of *Tsc2* is not a good strategy to treat sarcopenia.

Results

Premature death within a month after weaning was observed in a proportion of the mAkt/TscTKO mice (Fig. 7j), and even the survivors for 8 weeks after birth exhibited shortened longevity (Fig. 7j) with their dead body weight < 20 g (Supplementary Fig. 10m). (p.16, ll.350-353)

> Specific points:

> ·Fig 1C should be quantified. The mentioned reduction in insulin-stimulated signaling is not that convincing. Is it significant? If so for which targets?

Accordingly, we quantified insulin signaling 10 minutes after administration of insulin, and found that activation of at least FoxO1 and mTOR was significantly impaired in fast-twitch muscle of the mAktDKO mice (Fig. 1c & Supplementary Fig. 2c).

> ·Fig 3. The study of autophagy is not sufficient. They do not show flux measurements and the LC3 blot is of poor quality. They also mention mitophagy, but never show any on this.

> Fig 3B shows a lot of holes in the fibers of the ko. Are these artefacts or are these really present in the fibers?

In accordance with the suggestions by the reviewer, we examined autophagy flux by administration of lysosomal inhibitor, leupeptin, with improvement in western blotting of LC3 protein. LC3B-II protein was shown to be increased by lysosomal inhibition in the control floxed mice but it was not in the mAktDKO mice, suggesting the presence of impairment in autophagy flux.

As for mitophagy, we first isolated mitochondrial fraction and found that LC3B-II protein was detected in the fraction. Moreover, we quantified mitophagy-like events

showing autophagosomes containing abnormal mitochondria inside in electron microscopic analysis, and found that they were significantly increased in the mAktDKO mice.

We added descriptions, as follows:

Results

LC3B-II protein was increased by lysosomal inhibition by administration of leupeptin in the control floxed mice but was not in the mAktDKO mice, suggesting impairment in autophagy flux. Moreover, LC3B-II protein was detected in mitochondrial fraction (Supplementary Fig. 5d,e), and in electron microscopic analysis, mitophagy-like events involving autophagosomes containing abnormal mitochondria inside were significantly increased, with electron-dense aggregates left nearby (Fig. 3e & Supplementary Fig. 5a), suggesting that mitophagy failure could be induced in the fast-twitch muscle of the mAktDKO mice. (pp.9-10, ll.186-194)

Methods

Leupeptin (Wako) was diluted in phosphate-buffered saline, and administered intraperitoneally at the dose of 40 mg/kg of BW 4 hours before sacrifice. (p.26, ll.574-575)

To prepare the mitochondrial fractions, fresh tissues were homogenized in 0.25 M sucrose buffer, containing 10 mM Tris-HCl at pH 7.4 and 1 mM EDTA, followed by centrifugation at 900 g for 10 minutes. The supernatant was subjected to centrifugation at 6000 g for 10 minutes, and the precipitates were resuspended in the liver buffer. (p.32, ll.675-679)

The percentage of mitophagy-like events were calculated following a previous report. (p.34, ll.724-725)

Moreover, we repeated ATPase staining by improving the step of freezing, and the artefacts disappeared. Moreover, we showed staining at pH 10.43 in which type 1 myofibers are stained in gray in addition to staining at pH 4.11 in which type 1 myofibers are stained in black, to confirm relatively increased type 1 myofibers (Fig. 3b).

Methods

For ATPase staining, tissues were rapidly frozen in liquid nitrogen-cooled isopentane, and transverse serial sections were incubated with barbital acetate solution at pH 4.11 or 10.43, followed by incubation with ATP incubating solution. (p.33, ll.696-699)

> ·The lack of induction of atrogin1 and murf1 is somewhat surprising and should be explained. They could increase the number of FoxO3 genes they control for. This is also not sufficient to conclude that protein degradation is not involved, particularly as they show an increase in p62 and LC3.

We had showed in the original manuscript that expression of the key genes involved in ubiquitin-proteasome system and regulated by FoxO3, *Fbxo32* (Atrogin-1) and *Trim63* (MuRF1), was not affected by knocking out of Akt (Supplementary Fig. 4b). Besides, we performed transcriptome analysis of the fast-twitch muscle of the mAktDKO mice, to meet the questions raised by the reviewer. The affected pathways were involved in mitochondria and electron transport chain, in accordance with the results of RT-PCR. However, those involved in the ubiquitin-proteasome system were not affected in the mAktDKO mice (Supplementary Table 1), suggesting that FoxO3-targeted genes or E3 ligases, including Atrogin-1 and MuRF1, could not affected in this model.

We showed these additional results in the revised manuscript, as follows:

Results

Moreover, those encoding electron transport chain components (*Ndfua8*), which are known to be regulated by insulin signaling, as well as sirtuin 3, which regulates acetylation of the components, were down-regulated (Fig. 3a & Supplementary Fig. 4a). These changes in mitochondria-related gene expression were found to highlight the results of transcriptome analysis (GEO accession: [GSE199074]) of the fast-twitch muscle of the aged mAktDKO mice (Supplementary Table 1). (p.9, ll.171-177)

Of note, genes involved in proteasome-mediated protein degradation, which are known to be positively regulated by FoxO3, and pathways involved in proteasome and ubiquitination were shown to be unaffected by RT-PCR and transcriptome analysis (GSE199074), respectively (Supplementary Fig. 4b & Supplementary Table 1), suggesting that the reduced muscle mass observed in

the mAktDKO mice could not be largely attributed to increased protein degradation by the proteasome pathway. (p.10, ll.196-202)

Methods

RNA sequencing (pp.30-31, ll.640-653)

> Fig 6 would suggest that you can take out akt1, akt2, foxo1 and foxo4 from your muscles and all the rest seems normal. The suggested inhibition of foxo in sarcopenia seems a bit extreme, while they show there are no changes in numerous E3 ligases, most other groups find a close correlation between foxo activity and E3 ligase induction. This discrepancy needs to be explained.

As stated above, we performed transcriptome analysis of the fast-twitch muscle of the mAktDKO mice (Supplementary Table 1), which showed that expression of FoxO3-targeted genes or E3 ligases were generally unaffected in this model.

We referred to these results, as follows:

Results

Of note, genes involved in proteasome-mediated protein degradation, which are known to be positively regulated by FoxO3 27, and pathways involved in proteasome and ubiquitination were shown to be unaffected by RT-PCR and transcriptome analysis (GSE199074), respectively (Supplementary Fig. 4b & Supplementary Table 1), suggesting that the reduced muscle mass observed in the mAktDKO mice could not be largely attributed to increased protein degradation by the proteasome pathway. (p.10, ll.196-202)

Recently, FoxO1 and FoxO4 were reported to be the major FoxOs regulated by Akt to mediate proteostasis, although FoxO3 was rather regulated by the glucocorticoid pathway (Ref. #29, newly added in the revised manuscript), which seems consistent with our data.

Results

We focused on Foxo1 and Foxo4, major isoforms of FoxOs in skeletal muscle which were recently reported to be major FoxOs downstream of Akt 29, and generated skeletal-specific Akt1/Akt2/Foxo1/Foxo4 quadruple-knockout

(mAkt/FoxoQKO) mice (Supplementary Fig. 9a,b). (p.14, ll.305-308)

Moreover, we also showed the importance of inhibition of the FoxO pathway by administration of AS1842856, an inhibitor of FoxOs, especially FoxO1. Fast-twitch muscle mass of the mAktDKO mice at the age of 90 weeks was partially but significantly reversed by administration of the inhibitor daily for 4 weeks, in consistent with the reversal of phenotypes by additional knocking out of *Foxo1* and *Foxo4* in Fig. 6. Thus, it was shown that chemical inhibition of the FoxO pathway could be therapeutically effective against sarcopenia even in aged subjects. We added descriptions in the Results section and the Discussion section, as follows:

Results

Indeed, administration of a FoxO inhibitor, AS1842856 30, for 4 weeks partially reversed the reduced fast-twitch muscle mass of the aged mAktDKO mice, whereas it did not affect slow-twitch muscle mass (Fig. 6o & Supplementary Fig. 9i,j). (p.15, ll.323-326)

Discussion

Moreover, administration of a FoxO inhibitor, AS1842856 showed a partial but significant recovery of the reduced fast-twitch muscle mass of the mAktDKO mice. These data suggest that the FoxO pathway in skeletal muscle could be a good therapeutic target in this context, and it should be investigated in future works how to optimize the dosage and delivery of FoxO inhibitors, formally developed as anti-diabetic drugs, to prove their effectiveness against sarcopenia. (p.21, ll.465-471)

Methods

AS1842856, or 5-amino-7-(cyclohexylamino)-1-ethyl-6-fluoro-4-oxo-1,4-dihydroquinoline-3-carboxylic acid (Universal Biologicals), was diluted in 6% β -cyclodextrin (Sigma), and administered orally at the dose of 100 mg/kg of BW once daily for 4 weeks. (p.26, ll.577-580)

Responses to the comment made by the Reviewer 2:

We are extremely grateful to the Reviewer 2 for the very careful review of our manuscript and for the suggestions that made our manuscript markedly improved.

> Reviewer #2 (Remarks to the Author):

> Key results

> The goal of the present study was to determine whether disruption of the insulin/IGF-1 signaling by suppressing Akt activity in mouse skeletal muscle could accelerate sarcopenia and consequent systemic senescence by regulating anabolic and catabolic factors. The authors used Cre-LoxP system to generate skeletal muscle-specific Akt knockout mouse model and reported promoted sarcopenia and declines in grip strength and exercise endurance, accelerated osteopenia via impaired osteogenesis, induced insulin resistance, and premature death from debilitation. The underlying mechanisms were investigated by monitoring mitochondria-related parameters, and by genetically inactivating or activating the FoxO or mTOR pathway respectively. Their analyses reveal reduction in ATP content and abnormalities of mitochondria. In addition, data suggest that ablation of FoxOs can prevent sarcopenia, osteopenia, and premature death caused by Akt knockout, whereas activation of mTOR pathway has no major effect.

> Validity

> In general, the experimental designs, data collection, data analysis, and data interpretation were appropriate.

> Significance

> The overall scientific question is an important one and the results help to advance our understanding of how Akt activity in skeletal muscle impact muscle physiology and systemic senescence.

Thank you for the positive comments to our work.

> Data and methodology

> Results

> In general, the quality of presentation can be improved by rearranging some of the figures and adjusting the size of some images.

Accordingly, we rearranged and resized the images especially in Fig. 3.

> - Specific comments and concerns:

> 1. For the first subsection, in line 114-117, the authors made the statement that phosphorylation of these factors was suppressed to a lesser extent in slow-twitch muscle. Please quantify the signals to support the statement. Also, if you are looking at the differences in phosphorylation of a protein, e.g. AKT, it would be good to also show the expression of total Akt in order to distinguish between a change in AKT expression vs a change in the phosphorylation status of AKT.

We agree with the reviewer that quantification of expression or phosphorylation of proteins would contribute to deeper characterization of the knockout mice. Although we consider that the ratio of phosphorylated protein to an inner control (e.g. GAPDH) is important (Ref. #45), in accordance with the suggestion made by the reviewer, we examined p-Akt/Akt ratio and it was significantly lower in both fast-twitch muscle and slow-twitch muscle of the mAktDKO mice, but to a lesser extent in the latter, probably reflecting the knocking out efficiency and contamination of cells other than myocytes (Fig. 1c,d & Supplementary Fig. 2c,d).

We improved the manuscript, as follows:

Results

Knocking out of Akt was less sufficient in slow-twitch muscle and phosphorylation of the downstream molecules were not suppressed (Fig. 1d & Supplementary Fig. 2d). (p.7, ll.118-120)

> 2. For the second subsection, in line 148-153, the authors made the comparison between mAktDKO and Akt1/Akt2KO. However, Fig. S2i-l only showed data from Akt1/Akt2KO at the age of 32 weeks. According to Fig. 1d and Fig. 2f, both body weight and Ad libitum PG before insulin challenge had only little or even no difference from AktDKO mice to the control mice at the age of 40 weeks, thus, data from Fig. S2i-l (32 weeks) may not be supportive enough to make that statement.

As was pointed out by the reviewer, *ad libitum* plasma glucose was not affected in the mAktDKO mice, and we accordingly deleted data on plasma glucose (Supplementary Fig. 2j,l in the original manuscript) and just showed those on body

weight of skeletal muscle-specific *Akt1* or *Akt2* single-knockout mice in the revised manuscript, as follows:

Results

Therefore the mAktDKO mice can be a good model to elucidate the pathogenesis and mechanisms of sarcopenia, in which loss of mass and dysfunction of skeletal muscle were induced and accelerated with aging, accompanied by insulin resistance and glucose intolerance, in contrast to the skeletal muscle-specific *Akt1* or *Akt2* single-knockout mice, which did not exhibit reduced body weight (Supplementary Fig. 3h,i). (p.8, ll.152-157)

> 3. For the third subsection, in line 164-165 and line 188-189, it will guide the readers better if you can list out the genes.

Accordingly, we listed genes shown in Fig. 4, so that the readers could easily get the point, as follows:

Results

Genes highlighting fast-twitch muscle, type 2 myofibers (represented by *Myh4*) and glycolysis (*Pfkm*), were down-regulated in the fast-twitch muscle of the older mAktDKO mice (Fig. 3a & Supplementary Fig. 4a), while that encoding type 1 myofibers (*Myh7*) were not (Fig. 3a). (p.8, ll.162-165)

The other mechanism responsible for ATP production is oxidative phosphorylation, and genes involved in mitochondria biogenesis (*Tfam*, *Mfn2*) were down-regulated (Fig. 3a & Supplementary Fig. 4a), followed by a decrease in mitochondria DNA copy number (Fig. 3c). Moreover, those encoding electron transport chain components (*Ndufa8*), which are known to be regulated by insulin signaling, as well as sirtuin 3, which regulates acetylation of the components, were down-regulated (Fig. 3a & Supplementary Fig. 4a). (p.9, ll.168-174)

Autophagy-related genes (*Atg4a*) were found to be down-regulated (Fig. 3a & Supplementary Fig. 4a), but conversely, LC3B-II protein, a component of autophagosome, was accumulated (Fig. 3f). LC3B-II protein was increased by lysosomal inhibition by administration of leupeptin in the control floxed mice but was not in the mAktDKO mice, suggesting impairment in autophagy flux.

(p.9, ll.184-188)

Genes involved in reactive oxygen removal (*Sod2*), as well as in fatty acid oxidation, were also down-regulated (Fig. 3a & Supplementary Fig. 4a). (p.10, ll.194-195)

Given that mitochondria, oxidative stress, and proteostasis are closely associated with senescence, these data implied accelerated senescence in the mAktDKO mice, and indeed, sirtuins were down-regulated (Supplementary Fig. 4a), whereas a senescence marker (*Ink4a*) was highly up-regulated in the fast-twitch muscle of the mAktDKO mice (Fig. 3a), associated with enhanced staining of the senescence-associated beta-galactosidase (Fig. 3g). (p.10, ll.203-208)

> 4. For the fourth subsection, I have difficulty following the shift in focus. It will be better to explain how data in skeletal muscle promoted you to study senescence of other tissues.

Because the mAktDKO mice were shown to have trouble in mitochondria, oxidative stress, and proteostasis partly regulated by autophagy which are closely associated with senescence, we considered that senescence could be accelerated.

Moreover, of other tissues, skeletal muscle and bone are well known to be closely associated with each other, which prompted us to focus on osteoporosis, one of the phenomena associated with senescence in bone.

We explained these points in the revised manuscript with an additional reference (#28), as follows:

Results

Given that mitochondria, oxidative stress, and proteostasis are closely associated with senescence 28, these data implied accelerated senescence in the mAktDKO mice, and indeed, sirtuins were down-regulated (Supplementary Fig. 4a), whereas a senescence marker (*Ink4a*) was highly up-regulated in the fast-twitch muscle of the mAktDKO mice (Fig. 3a), associated with enhanced staining of the senescence-associated beta-galactosidase (Fig. 3g). (p.10, ll.203-208)

These data prompted us to hypothesize that accelerated senescence of skeletal muscle could affect senescence of other tissues, and even systemic senescence.

Because skeletal muscle and bone are closely associated with each other, we focused on osteoporosis, one of the phenomena associated with senescence in bone. (p.10, ll.213-217)

> 5. Fig. 1c, as stated above, it would be better to have quantification of the signals and western blots on total proteins. In Fig1c and supplemental Fig2c, the substrate protein levels should also be assayed to insure the significant changes of phosphorylation.

As stated above, in accordance with the suggestion made by the reviewer, we quantified phosphorylation of key downstream molecules, FoxO1, mTOR, and AS160, in comparison with the total protein. It was impaired in fast-twitch muscle by ablation of Akt, but it was not markedly affected in slow-twitch muscle, again probably reflecting the knocking out efficiency of Akt (Fig. 1c,d & Supplementary Fig. 2c,d). We improved the manuscript, as follows:

Results

Knocking out of Akt was less sufficient in slow-twitch muscle and phosphorylation of the downstream molecules were not suppressed (Fig. 1d & Supplementary Fig. 2d). (p.7, ll.118-120)

> 6. Fig. 2c and 2e, labels for the x-axis or y-axis respectively would help the readers to understand the figure.

We accordingly arranged the labels in Fig. 2c,e.

> 7. Fig. 3f, bands for LC3BI and LC3BII are not apparent or clear on this blot. Also, please label the positions clearly.

We improved blotting of LC3B protein and clearly showed LC3B-I and LC3B-II proteins with molecular weight indicators in Fig. 3f & Supplementary Fig. 5e.

> 8. Fig. 4, please use the correct format of the units. For example, please use mm³ instead of mm3.

We accordingly superscripted numbers in Fig. 4,6,7 & Supplementary Fig. 6,9.

> 9. In supple Fig7b, the substrate protein levels should be provided.

We accordingly added total Akt and total mTOR in now Supplementary Fig. 10b.

> 10. Fig. S2c, blot for p-S6K is missing.

As stated above, we focused on phosphorylation of FoxO1, mTOR, and AS160, and quantified phosphorylated protein-to-total protein ratios both in fast-twitch muscle and slow-twitch muscle (Fig. 1c,d & Supplementary Fig. 2c,d).

> 11. Akt(Thr308) is the positive upstream regulator of mTORC1. Can the authors explain the description in line 303-304? Many descriptions of Akt signaling pathway are inaccurate.

As was stated by the reviewer, Akt positively regulates mTORC1, and thus mTORC1 was inactivated by ablation of Akt (Fig. 1c). That is why we tried to activate mTORC1 in order to reverse the phenotypes of the mAktDKO mice.

> 12. TSC1/2 can regulate biological processes through mTOR-dependent or -independent pathways, the potential role of mTOR-independent regulation should be discussed.

We accordingly add discussion on this point, as follows:

Results

Although knocking out of Tsc2 could also affect the mTOR-independent pathways, which might have affected the phenotypes of the mAkt/TscTKO

mice, overall, activation of the mTOR pathway in skeletal muscle was not considered a good strategy for treating sarco-osteopenia. (p.20, ll.455-458)

> Methods

> 1. In exercise tolerance test, please provide reference(s) for the chosen parameters. Also, please explain why 15 min was chosen to measure the ATP content.

We chose extended duration of exercise, 60 minutes, in the younger mice, because exercise duration examined during 15 minutes of exercise (Ref. #52) was not impaired in the mAktDKO mice at the age. On the other hand, we chose 15 minutes in the older mice, just as in assessment of exercise duration, because exercise duration was impaired, and they could not run any longer. We added description, as follows:

Methods

In exercise tolerance test, mice were subjected to running on the treadmill without electrical shocks at a speed of 15 m/minute with a gradient of 14 degrees for 900 seconds in the older mice, just as in assessment of exercise duration, and for extended duration of 3600 seconds in the younger mice. (p.25, ll.554-557)

Results

Besides, the exercise duration was markedly shorter in the older mAktDKO mice (Fig. 2b), and consistently, the ATP content in skeletal muscle of the mAktDKO mice after exercise was decreased compared to that of the control mice, although it was not even after extended endurance exercise in the younger mAktDKO mice (Fig. 2c). (p.7, ll.137-141).

> 2. For “survival” section, please provide reference for the use of 20 g in the judgement of debilitation.

As stated in the Results section, we defined deaths from debilitation as those with no macroscopic findings and with a dead body weight less than 20 g, which was equivalent to the 10 percentile of body weight of the control mice, as shown in Fig.

4h. We added description to the Methods section, as follows:

Methods

Those without any macroscopic findings and with dead body weight less than 20 g, equivalent to the 10 percentile of body weight of the control mice, were judged to have died from debilitation. (pp.25-26, ll.566-568)

> 3. For “gene expression analysis”, please provide company information for RNeasy Mini Kit.

Accordingly, we showed that the kit as well as Proteinase K were provided by Qiagen.

> 4. For “primers” section, why are the primers sequences in both lowercase and uppercase?

Accordingly, we showed all the sequences in uppercase letters.

> Analytical approach

> In general, the statistical methods are valid and correctly applied. With that said, it would be more appropriate to use Chi-square instead of t-test in the analysis of CSA distribution.

Accordingly, we used Chi-square test and confirmed that the difference was statistically significant in Fig. 1h.

> Clarity and context

> Overall, the manuscript would benefit for a thorough review for grammar, misspelled words, and odd sentence structure.

We had the manuscript reviewed and improved it accordingly.

> References

> 1. In line 89 and 90, please add reference(s) for streptozotocin treatment and db/db mice.

We inserted Ref. #23, as follows:

Results

To elucidate the effects of insulin signaling on skeletal muscle mass, we first investigated two mouse models, i.e., mice treated with streptozotocin, a model of insulin deficiency, and *db/db* mice, a model of severe insulin resistance 23.

Methods

C57BL/6J mice were purchased from Oriental Yeast, and BKS.Cg-+ *Lepr^{db/+}* *Lepr^{db}/Jcl* (*db/db*) mice from CLEA Japan 23.

Streptozotocin (SIGMA) was diluted in citrate sodium buffer, and administered intraperitoneally at the dose of 150 mg/kg of body weight (BW) twice with an interval of 4 days 23.

> 2. In line 282 and 283, please add reference(s) for the statements on FoxO and mTOR.

We inserted Ref. #9 & #10.

> 3. In line 304-305, please add reference(s) for the statement on TSC2.

We inserted Ref. #9 & #10.

Responses to the comment made by the Reviewer 3:

We are extremely grateful to the Reviewer 3 for the very careful review of our manuscript and for the suggestions that made our manuscript markedly improved.

> Reviewer #3 (Remarks to the Author):

> Sasako et al. present an interesting manuscript investigating the role of Akt in skeletal muscle to protect against sarcopenia and senescence. The present interesting evidence indicating that suppression of Akt1/2 in skeletal muscle associated with insulin resistance and aging could accelerate sarco-osteopenia and systemic senescence.

> Hypothesis is that sarcopenia could be promoted by insulin resistance in skeletal muscle. This is interesting, but the more translatable question would be does increasing insulin sensitivity negate the sarcopenic phenotype?

Because most of the insulin sensitizers exert the effects through Akt downstream, in order to treat insulin resistance in muscle of the mAktDKO mice, we selected administration of AS1842856, an inhibitor of FoxOs, especially FoxO1, not other popular insulin sensitizers. Indeed, fast-twitch muscle mass of the mAktDKO mice at the age of 90 weeks was partially but significantly reversed by administration of the inhibitor daily for 4 weeks, in consistent with the reversal of phenotypes by additional knocking out of *Foxo1* and *Foxo4* in Fig. 6. Thus, it was shown that chemical inhibition of the FoxO pathway that should be also suppressed by insulin could be therapeutically effective against sarcopenia even in aged subjects.

We added descriptions in the Results section and the Discussion section, as follows, with a reference on another FoxO1 inhibitor (Ref. #40 in the original manuscript; Eur J Pharmacol 645: 185-191, 2010) deleted:

Results

Indeed, administration of a FoxO inhibitor, AS1842856 30, for 4 weeks partially reversed the reduced fast-twitch muscle mass of the aged mAktDKO mice, whereas it did not affect slow-twitch muscle mass (Fig. 6o & Supplementary Fig. 9i,j). (p.15, ll.323-326)

Discussion

Moreover, administration of a FoxO inhibitor, AS1842856 showed a partial but significant recovery of the reduced fast-twitch muscle mass of the

mAktDKO mice. These data suggest that the FoxO pathway in skeletal muscle could be a good therapeutic target in this context, and it should be investigated in future works how to optimize the dosage and delivery of FoxO inhibitors, formally developed as anti-diabetic drugs, to prove their effectiveness against sarcopenia. (p.21, ll.465-471)

Methods

AS1842856, or 5-amino-7-(cyclohexylamino)-1-ethyl-6-fluoro-4-oxo-1,4-dihydroquinoline-3-carboxylic acid (Universal Biologicals), was diluted in 6% β -cyclodextrin (Sigma), and administered orally at the dose of 100 mg/kg of BW once daily for 4 weeks. (p.26, ll.577-580)

Moreover, we also treated the mAktDKO mice at the age of 8 weeks with AS1842856 for 4 weeks and also found that fast-twitch muscle mass was partially but significantly reversed, suggesting that inhibition of the FoxO pathway could be effective against prevention of sarcopenia as well. Please see Fig. 1 Only for the Reviewers attached below.

> Did the authors look at the effects of exercise in the muscle of the older mAktDKO mice? After the acute exercise bout – would be of interest to determine if other signaling pathways (i.e. Ampk / pAmpk) are similarly altered with age or still able to be stimulated with exercise, even in the absence of Akt.

Accordingly, we examined activation of AMPK after exercise in the younger and the older mAktDKO mice. At the age of 8 weeks, when exercise endurance was not impaired, phosphorylation of AMPK α was significantly elevated in the fast-twitch muscle of the mAktDKO mice even after extended exercise for 60 minutes (Supplementary Fig. 5b), in consistent with the ATP content which was not decreased in the mAktDKO mice (Fig. 2c). However, at the age of 24 weeks, when exercise endurance was not impaired, phosphorylation of AMPK was not elevated (Supplementary Fig. 5c), in consistent with the ATP content which was significantly decreased in the mAktDKO mice (Fig. 2c). Thus, activation of the AMPK pathway was observed only in the younger mAktDKO mice, probably in a compensatory manner, but it was declined with aging, which could contribute to the development of impaired exercise duration observed in the older mAktDKO mice, and the

mechanisms underlying the decline of AMPK pathway should be explored in future works.

We added descriptions in the Results section Discussion, as follows:

Results

AMPK α was activated even after extended endurance exercise in the younger mAktDKO mice but it was not in the older mAktDKO mice (Supplementary Fig. 5b,c), in accordance with the changes in ATP content (Fig. 2c). (p.9, ll.180-182)

Discussion

Most of these phenotypes associated with knocking out of Akt in skeletal muscle appeared at the age of 8 weeks or much later, possibly due to some compensatory mechanisms that might deteriorate with aging, and at least mitochondrial dys-function was shown likely to be compensated by activation of the AMPK pathway in the younger mAktDKO mice. (p.18, ll.405-409)

> For Fig. 4, why is the n only 3? An n=5-6 would provide greater support to the conclusions.

Accordingly, we performed additional experiments and confirmed that bone mineral density was significantly decreased in spongy bone, but not in cortical bone, in the mAktDKO mice (n = 6) in Fig. 4b. It suggests that some myokines or metabolites derived from skeletal muscle regulated by Akt, rather than changes in mechanical loading, could affect osteogenesis, as was discussed in the Discussion section.

Results

Computed tomography (CT) scanning of lower leg revealed decreased bone mineral density (BMD) in spongy bone compared to the control mice (Fig. 4b), and the star volume, a good parameter of osteoporosis, was larger in the femur of the mAktDKO mice on micro-CT scanning (Fig. 4c & Supplementary Fig. 6a). (p.11, ll.218-222)

Discussion

In addition to sarcopenia, the mAktDKO mice exhibited bone loss, which we call

‘sarco-osteopenia’. Reduced mechanical loading is among its potential mechanisms, given that the mAktDKO mice exhibited decreased muscle mass and impaired motor function. However, it is known that cortical bones are mainly affected by mechanical loading, whereas spongy bones were mainly affected in the mAktDKO mice. Moreover, osteopenic changes were observed before muscle loss became evident. These data suggest that some myokines or metabolites derived from skeletal muscle regulated by Akt (and presumably through FoxOs) could affect osteogenesis, which should be explored and elucidated in future studies. (p.18, ll.396-404)

> The authors mentioned that these experiments were performed in male and female mice, but it is not clear throughout if the data was pooled or is presented in male vs. female mice.

All the experiments were performed using male mice, except those for Supplementary Fig. 7 to examine phenotypes of the female mAktDKO mice. We added a sentence to the Methods section, as follows:

Methods

Male mice were subjected to experiments, unless otherwise indicated. (p.23, ll.501-502)

Fig. 1 Only for the Reviewers

Body weight at baseline and skeletal muscle weight of the mAktDKO mice at the age of 8 weeks treated with AS1842856 at the dose of 100 mg/kg of BW daily for 4 weeks (n = 3-4). bCD: β -cyclodextrin, AS: AS1842856.

REVIEWERS' COMMENTS

Reviewer #1 (Remarks to the Author):

The authors improved the manuscript in the revision.

I continue to not understand why the authors insist on the senescence interpretation. They show mice lacking Akt1/2 in skeletal muscle die prematurely, but this is very likely due to muscle defects as evidenced by the strong reduction in exercise capacity. It is very plausible that mice, in which you reduce/eliminate important proteins from muscle, there is a performance issue which can lead to premature death. They should reduce this senescence interpretation throughout the manuscript.

Reviewer #2 (Remarks to the Author):

The authors have responded to most of my comments. However, as stated in the previous peer review, the quality of the presentation can be improved by rearranging figures and images. In addition, please pay attention to the labeling in all figures. Some of the mislabeling and typos are listed in the "specific comments" below. But please go over the manuscript again to catch all of them.

Specific comments:

1. In Supplementary Fig. 2, both c and d were labelled as "gastrocnemius". In Fig. 1d, typo "solues".

1. In Supplementary Fig. 2c and 2d, the graphs for quantification of "Akt/GAPDH" and "p-Akt/Akt" should be placed next to the specific blots.

2. In Fig. 2e, y-axis are still not labelled.

3. Top blot in Fig. 3f, the label on the right says "p-Akt", while the labels on the left are "LC3B-I" and "LC3B-II". Bottom blot in Fig. 3f, I think the blot labelled as "Akt" should be for "Gapdh".

Reviewer #3 (Remarks to the Author):

All my comments have been addressed.

Responses to the Comments Made by the Reviewers

Responses to the comment made by the Reviewer 1:

We are extremely grateful to the Reviewer 1 for further suggestions.

> Reviewer #1 (Remarks to the Author):

> The authors improved the manuscript in the revision.

> I continue to not understand why the authors insist on the senescence interpretation. They show mice lacking Akt1/2 in skeletal muscle die prematurely, but this is very likely due to muscle defects as evidenced by the strong reduction in exercise capacity. It is very plausible that mice, in which you reduce/eliminate important proteins from muscle, there is a performance issue which can lead to premature death. They should reduce this senescence interpretation throughout the manuscript.

In accordance with the suggestions by the reviewer, we modified the title, the main text, and Figure 9 to show our schematic summary, mainly by using “shortened longevity” or “premature death” instead of “systemic senescence”, as follows:

Title

Role of Akt in skeletal muscle in protection against sarcopenia and maintenance of longevity (p.1, ll.1-2, in a clean copy hereafter)

Abstract

Overall, our data suggest that, unlike in lower organisms, suppression of Akt activity in skeletal muscle of mammals associated with insulin resistance and aging could accelerate sarco-osteopenia and consequently shortened longevity. (p.3, ll.38-41)

Main Text (Introduction)

They also showed osteopenia and shortened longevity, suggesting a crucial role for skeletal muscle Akt in the regulation of senescence of skeletal muscle and longevity. (p.5, ll.83-85)

Results

These data prompted us to hypothesize that accelerated senescence of skeletal muscle could affect senescence of other tissues, ~~and even systemic senescence.~~

(p.10, ll.215-216)

The cumulative incidence of death from debilitation showed an apparent separation, but that of death from tumor did not (Supplementary Fig. 6g).

~~Therefore it was shown that sarcopenia accelerated systemic senescence, resulting in shortened systemic longevity, even when associated with changes of causes of death.~~ (p.10, ll.246-248)

Discussion

Here we show, rather, that impaired insulin/IGF-1 signaling in skeletal muscle in mice not only accelerates senescence of skeletal muscle itself but also shortens longevity. (p.12, ll.363-365)

It is noteworthy that knocking out of just a single kinase in just a single tissue not only mildly affects longevity but even drastically affects the cause of death. (p.17, ll.417-419)

These changes may be amenable to nearly complete reversal by inactivation of FoxOs, which are thus thought to represent a likely therapeutic target in coping with skeletal muscle-centered ~~systemic~~ senescence. (p.19, ll.488-490)

Figure 9

Systemic senescence -> Osteopenia & premature death

Responses to the comment made by the Reviewer 2:

We are extremely grateful to the Reviewer 2 for further suggestions.

> Reviewer #2 (Remarks to the Author):

> The authors have responded to most of my comments. However, as stated in the previous peer review, the quality of the presentation can be improved by rearranging figures and images. In addition, please pay attention to the labeling in all figures. Some of the mislabeling and typos are listed in the “specific comments” below. But please go over the manuscript again to catch all of them.

In accordance with the suggestion made by the reviewer, we split the former Figure 1 into two figures, Figure 1 in the revised manuscript to show generation of skeletal muscle-specific Akt knockout mice and Figure 2 in the revised manuscript to show body weight and muscle weight of the knockout mice.

We modified the mislabeling and typos kindly pointed out by the Reviewer 2, as follows.

> Specific comments:

> 1. In Supplementary Fig. 2, both c and d were labelled as “gastrocnemius”. In Fig. 1d, typo “solues”.

“Soleus” is correct both in Supplementary Fig. 2d and Fig. 1d.

> 1. In Supplementary Fig. 2c and 2d, the graphs for quantification of “Akt/GAPDH” and “p-Akt/Akt” should be placed next to the specific blots.

Accordingly, now we show the corresponding graphs in Fig. 1c and Fig. 1d.

> 2. In Fig. 2e, y-axis are still not labelled.

Now the y-axis of Fig. 3e (formerly Fig. 2e) is labeled “2-DG uptake [nmol/g/min]”.

> 3. Top blot in Fig. 3f, the label on the right says “p-Akt”, while the labels on the left are “LC3B-I” and “LC3B-II”. Bottom blot in Fig. 3f, I think the blot labelled as “Akt” should be for “Gapdh”.

Now the blots are properly labeled “LC3-B” and “GAPDH” respectively in Fig.4f (formerly Fig. 3f).

Responses to the comment made by the Reviewer 3:

- > Reviewer #3 (Remarks to the Author):
- > All my comments have been addressed.

We really appreciate the comment made by the Reviewer 3.